# Phospho-regulated Bim1/EB1 interactions trigger Dam1c ring assembly at the budding yeast outer kinetochore

Alexander Dudziak[1] (ID), Lena Engelhard[2], Cole Bourque[2,3], Björn Udo Klink[2,3], Pascaline Rombaut[4], Nikolay Kornakov[1] (ID), Karolin Jänen[1], Franz Herzog[4] (ID), Christos Gatsogiannis[2,3] & Stefan Westermann[1,*] (ID)

## Abstract

Kinetochores form the link between chromosomes and microtubules of the mitotic spindle. The heterodecameric Dam1 complex (Dam1c) is a major component of the *Saccharomyces cerevisiae* outer kinetochore, assembling into 3 MDa-sized microtubule-embracing rings, but how ring assembly is specifically initiated *in vivo* remains to be understood. Here, we describe a molecular pathway that provides local control of ring assembly during the establishment of sister kinetochore bi-orientation. We show that Dam1c and the general microtubule plus end-associated protein (+TIP) Bim1/EB1 form a stable complex depending on a conserved motif in the Duo1 subunit of Dam1c. EM analyses reveal that Bim1 crosslinks protrusion domains of adjacent Dam1c heterodecamers and promotes the formation of oligomers with defined curvature. Disruption of the Dam1c-Bim1 interaction impairs kinetochore localization of Dam1c in metaphase and delays mitosis. Phosphorylation promotes Dam1c-Bim1 binding by relieving an intramolecular inhibition of the Dam1 C-terminus. In addition, Bim1 recruits Bik1/CLIP-170 to Dam1c and induces formation of full rings even in the absence of microtubules. Our data help to explain how new kinetochore end-on attachments are formed during the process of attachment error correction.

**Keywords** Bik1; Bim1; bi-orientation; chromosome segregation; Dam1/DASH complex

**Subject Categories** Cell Cycle; Post-translational Modifications & Proteolysis

**The EMBO Journal (2021) 40: e108004**

## Introduction

Kinetochores assembled on centromeric DNA constitute the physical linkage between chromosomes and microtubules of the mitotic spindle. Error-free chromosome segregation requires the formation of bi-oriented end-on kinetochore–microtubule attachments. Since other attachment configurations result in chromosome mis-segregation, cells have evolved mechanisms to correct erroneous attachments (Biggins, 2013). In the presence of incorrect attachments, characterized by the inability to sustain tension, activity of the conserved Ipl1/Aurora B kinase disrupts kinetochore–microtubule attachments by phosphorylation of the Dam1 (Dam1c) and Ndc80 complexes (Ndc80c). Phosphorylation of these microtubule-binding complexes weakens their affinity for microtubules and for each other (Wang *et al*, 2007; Lampert *et al*, 2010; Tien *et al*, 2010). Generating unattached kinetochores finally results in the activation of the spindle assembly checkpoint (SAC), which prevents anaphase onset until all kinetochores are attached properly (Musacchio, 2015). Error correction requires repeated attempts of formation and release of end-on attachments. However, while the mechanisms of release have been studied in detail, activities promoting the formation of end-on attachments are less well characterized. Even though microtubule plus ends provide a much smaller interaction surface compared with the lateral microtubule lattice, end-on attachments are the only configuration that allows faithful chromosome segregation. Since not all newly established end-on attachments will immediately sustain tension, they need to be regulated and released in an Ipl1-dependent manner unless they are stabilized by so far poorly understood mechanisms.

One of the main microtubule-binding complexes of the kinetochore of *Saccharomyces cerevisiae* is the heterodecameric Dam1 complex (Dam1c). The Dam1 subunit of this complex is a key substrate of the Ipl1/Aurora B kinase in the context of error correction (Cheeseman *et al*, 2002). Furthermore, Dam1c has the unique ability to oligomerize into rings, which can completely encircle microtubules (Miranda *et al*, 2005; Westermann *et al*, 2005, 2006; Jenni & Harrison, 2018; Ng *et al*, 2019). Formation of the Dam1c ring, which is thought to attach to the kinetochore via the Ndc80 complex (Ndc80c) and can slide along depolymerizing

1   Department of Molecular Genetics I, Center of Molecular Biotechnology, University of Duisburg-Essen, Essen, Germany
2   Department of Structural Biochemistry, Max Planck Institute of Molecular Physiology, Dortmund, Germany
3   Institute for Medical Physics and Biophysics and Center for Soft Nanoscience, Westfälische Wilhelms-Universität Münster, Münster, Germany
4   Gene Center Munich, Ludwig Maximilian University Munich, Munich, Germany
    *Corresponding author. Tel: +49 2011832733; E-mail: stefan.westermann@uni-due.de

microtubules, has been described as suitable tool to couple microtubule dynamics to chromosome movement (Westermann *et al*, 2006; Ramey *et al*, 2011). In addition, Dam1c strengthens binding of Ndc80c and isolated native kinetochores to dynamic microtubule ends against tension and promotes binding of Ndc80c to microtubules (Tien *et al*, 2010; Sarangapani *et al*, 2013; Kim *et al*, 2017).

While recombinant Dam1c forms rings along the entire microtubule lattice *in vitro*, Dam1c localizes predominantly to kinetochores *in vivo* and only weakly to other spindle microtubules. How Dam1c ring formation is regulated and restricted to kinetochores *in vivo* is unclear (Westermann *et al*, 2005; Ng *et al*, 2019). Thus, it is conceivable that this process is regulated in a precise manner, for instance, by kinetochore-localized kinases, in order to prevent precocious ring assembly. Previous studies have demonstrated that additional non-kinetochore proteins can make important contributions to the correct functionality of the kinetochore. Among these proteins are microtubule-associated proteins (MAPs) such as Stu2/ch-TOG and EB1 (Miller *et al*, 2016, 2019; Thomas *et al*, 2016). In addition, the autonomous microtubule plus end-tracking protein (+TIP) Bim1 (the yeast homolog of EB1) was identified as potential binding partner of Dam1c in comprehensive two-hybrid studies (Wong *et al*, 2007). Furthermore, genetic interactions between Bim1 and several kinetochore proteins identified in a large-scale screen suggest an important role for Bim1 during formation of kinetochore–microtubule attachments (Tong *et al*, 2001). Bim1 localizes to microtubule plus ends (Tirnauer *et al*, 1999) and is considered as master regulator of the microtubule network by regulating microtubule dynamics and recruiting additional proteins to plus ends (Akhmanova & Steinmetz, 2008, 2015). Besides MAPs, the conserved effector kinase of the spindle assembly checkpoint Mps1 contributes to the formation of kinetochore–microtubule attachments (Shimogawa *et al*, 2006; Maure *et al*, 2007). It was shown that Mps1 kinase activity is required for coupling kinetochores to microtubule plus ends (Shimogawa *et al*, 2006) and to promote bi-orientation (Maure *et al*, 2007); however, the molecular mechanisms that underlie these important Mps1 functions are not well understood.

Here, we combine biochemical reconstitution experiments with structural analysis and genetics to elucidate the mechanism of regulated Dam1c ring assembly at the yeast kinetochore. Phospho-regulated formation of a Dam1c-Bim1-Bik1 complex promotes oligomerization with stable, fully formed Dam1c rings as the result of the assembly process. We show that this Bim1-dependent pathway acts in parallel with Cdk1-dependent phosphorylation of the Ask1 subunit to control Dam1c oligomerization *in vivo*. Our experiments further reveal how kinetochore-localized Mps1 kinase opposes the activity of Ipl1 kinase and actively promotes the formation of new microtubule attachments via stepwise maturation through distinct outer kinetochore configurations.

## Results

### Dam1c and Bim1 bind in solution and form a stable complex

The autonomous plus end-tracking protein Bim1 was previously identified as a potential binding partner of the Dam1c subunit Duo1 by yeast two-hybrid screens (Wong *et al*, 2007). However, studies analyzing this interaction and its contribution to kinetochore functionality in cells are missing so far. We purified recombinant Dam1c and Bim1 from *Escherichia coli* and tested for *in vitro* complex formation by size-exclusion chromatography (Fig 1A). Addition of Bim1 to Dam1c resulted in a stoichiometric coelution of Bim1 with Dam1c. Previous studies have suggested an interaction between Dam1c and Stu2, which was identified by yeast two-hybrid studies (Wong *et al*, 2007; Kalantzaki *et al*, 2015). Like Bim1, Stu2 is a microtubule plus end-associated protein, which was previously described to make an important contribution to faithful chromosome segregation (Miller *et al*, 2016, 2019). We tested for binding of Dam1c to Stu2, which was purified from Sf9 insect cells. In contrast to Bim1, Stu2 did not show any physical interaction with Dam1c in size-exclusion chromatography and pull-down assay (Fig EV1A and B). This result shows that Bim1 is a specific binding partner of Dam1c.

To gain a deeper insight into the molecular topology of the novel Dam1c-Bim1 complex, Dam1c was treated with the lysine-specific crosslinker BS3 (bis(sulfosuccinimidyl)suberate) in the absence and presence of Bim1 and analyzed by mass spectrometry. Multiple crosslinks between the individual subunits of Dam1c were identified in the absence of Bim1 (Fig EV1C, Dataset EV1). Particularly, a high number of crosslinks between lysine residues located in the subunits Dam1, Duo1, Spc19, and Spc34 were mapped. Our crosslinking data are in agreement with previously published proximity maps of Dam1c in the absence or presence of microtubules (Zelter *et al*, 2015; Legal *et al*, 2016). The presence of Bim1 did not alter the overall pattern of crosslinks between the Dam1c subunits (Figs 1B and EV1C and Dataset EV1). Several crosslinks between Bim1 and

---

**Figure 1. Bim1 binds to Dam1c to form a stable complex in solution.**

A Chromatogram and SDS–PAGE analysis of elution fractions from analytical size-exclusion chromatography. Dam1c and Bim1 individually or in combination were subjected to gel filtration. Bim1 coelutes with Dam1c and forms a stoichiometric complex.

B Lysine–lysine proximity map of Dam1c-Bim1 complex crosslinked by BS3. Each protein is represented as bar. Lines between the different proteins represent a pair of crosslinked lysine residues identified by mass spectrometry. Subunits of Dam1c that were crosslinked with Bim1 are shown in red, all other subunits in blue and Bim1 in green. The positions of an SxIP sequence in the C-terminus of Duo1 and Q205 in the C-terminus of Dam1 are marked by vertical lines.

C Comparative structural analysis of representative 2D class averages of Dam1$^{WT}$c (upper row) and Dam1$^{WT}$c-Bim1 (lower row). Densities exclusive to Dam1$^{WT}$c (arrow) indicate the topology of Bim1 (scale bar: 10 nm).

D Three-dimensional structures of Dam1$^{WT}$c (blue) and Dam1$^{WT}$c-Bim1 (yellow) computed from the 2D class averages. The location of the protrusion domains of the Dam1$^{WT}$ complex (composed of Spc19/Spc34) is indicated (gray dashed ovals). Additional density in the Dam1$^{WT}$c/Bim1 reconstruction (yellow dashed oval) indicates binding of Bim1 to the protrusion domains of Dam1c.

E, F EM images of negatively stained Dam1c alone (E) and in the presence of Bim1 (F), at low salt conditions. The final concentrations of Dam1c and Bim1 were 0.3 and 0.1 µM, respectively. The scale bars are 100 nm, boxes mark representative complexes shown as magnified inserts.

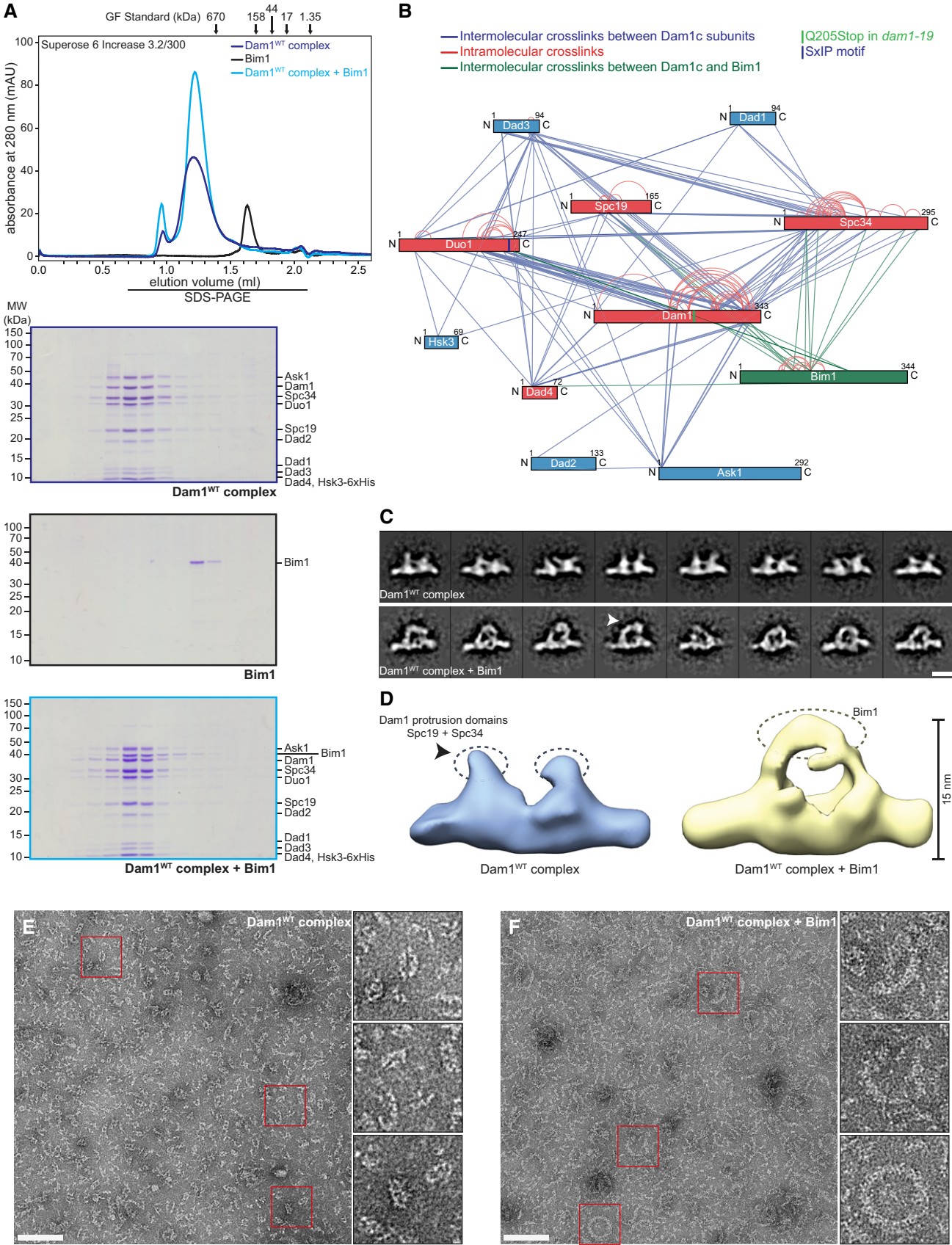

**Figure 1.**

five Dam1c subunits, Dam1, Duo1, Spc34, Spc19, and Dad4, were identified (Appendix Table S1). Interestingly, a high density of crosslinks was found between Bim1 and the C-termini of Duo1 and Dam1. These crosslinks are of particular interest for exploring the potential binding mechanism between Dam1c and Bim1, which will be characterized in the following experiments. Crosslinks between Bim1 and subunits of Dam1c are listed in Appendix Table S1.

## Single-particle EM analysis reveals binding of Bim1 in close proximity to the protrusion domain of Dam1c

To extend our knowledge of how Bim1 binds to Dam1c, we visualized the Dam1c-Bim1 complex by negative-stain electron microscopy. A stoichiometric Dam1c-Bim1 complex was purified by size-exclusion chromatography after crosslinking with glutaraldehyde to prevent dissociation of the complex. Consistent with previous studies (Wang *et al*, 2007), Dam1c predominantly appeared as a T-shaped monomer or a $\pi$-shaped homodimer of two heterodecamers in the absence of Bim1 (Figs 1C and D, and EV1D). Notably, both protrusion domains of a homodimer are aligned in the same direction emanating from the central part of the crossbar. Upon combination, Bim1 was visible as an additional mass crowning the protrusion domains of adjacent heterodecamers (Figs 1C and D, and EV1E). The size of the extra mass is estimated to accommodate at least one homodimeric Bim1 molecule (76.6 kDa). As the protrusion domains are formed by the subunits Spc19 and Spc34 (Jenni & Harrison, 2018), these findings are in full agreement with our crosslinking data, which identified crosslinks between Bim1 and Spc19/Spc34 and between Duo1 and Spc19/Spc34 (Fig 1B, Appendix Table S1 and S2).

As Bim1 appears to bridge the protrusion domains of two adjacent Dam1c monomers, we further investigated the effect of Bim1 binding on the oligomerization status of Dam1c by electron microscopy. To do this, we visualized Dam1c and Dam1c-Bim1 complex at low salt concentrations to support complex formation. We found that Dam1c appeared as monomers or small oligomeric species without forming ring-like structures (Fig 1E). In contrast, the Dam1c-Bim1 complex showed larger oligomers than Dam1c alone, with a curvature fitting to that of complete rings. In addition, a sparse formation of larger ring-like structures and even completely formed rings was observed (Fig 1F). We conclude that association between Bim1 and Dam1c can promote oligomerization as a step toward Dam1c ring formation at the kinetochore.

## An SxIP motif in the C-terminus of Duo1 is required for binding of Bim1 to Dam1c

End-binding (EB) proteins such as Bim1 share highly conserved domains that fulfill distinct functions in the context of microtubule and cargo binding. The N-terminal calponin homology (CH) domain binds to microtubules and is followed by a flexible linker and a coiled-coil region, which mediates homodimerization. EB proteins bind to cargo proteins either via an end-binding homology (EBH) domain close to the C-terminus or an EEY/F motif (ETF in case of Bim1) at the very C-terminus. EBH domain and ETF motif mediate binding to SxIP and CAP-Gly domains, respectively (Weisbrich *et al*, 2007; Honnappa *et al*, 2009). To assess the role of the different functional domains of Bim1 for binding to Dam1c, GST-tagged Bim1

truncations were purified from *E. coli* and used in pull-down assays with recombinant $Dam1^{WT}c$ (Appendix Fig S1A). Only GST-Bim1 constructs containing the full-length EBH domain were able to bind $Dam1^{WT}c$, and even a short N-terminal truncation of the EBH domain was sufficient to prevent binding. Interestingly, GST-Bim1 constructs containing only the EBH domain ($GST-Bim1^{185-344}$) displayed enhanced binding compared with the full-length protein (Appendix Fig S1A and B).

The EBH domain of Bim1 specifically recognizes and binds to SxIP (serine–any amino acid–isoleucine–proline) tetrapeptide sequences of their cargo proteins (Honnappa *et al*, 2009). A multiple sequence alignment of Duo1 proteins from various yeast species revealed a highly conserved SxIP motif in the unstructured C-terminus of the protein (amino acids 225–228; Fig 2A). Noticeably, $Duo1^{K236}$ was crosslinked with $Bim1^{K223}$, which is located in its EBH domain (Fig 1B, Appendix Table S1). To test whether the Duo1 SxIP motif is required for binding of Bim1, we purified recombinant Dam1c in which the Duo1 SxIP motif is deleted ($Dam1^{\Delta SxIP}c$). Bim1 failed to coelute with $Dam1^{\Delta SxIP}c$ in size-exclusion chromatography, and binding was also abolished in a solid-phase binding assay (Fig 2B and C). In contrast, deletion of these four amino acids did not change the elution position of $Dam1^{\Delta SxIP}c$ in size-exclusion chromatography and had no effect on complex integrity. To validate our findings, we constructed yeast strains expressing 6xFlag-tagged $Duo1^{WT}$ or $Duo1^{\Delta SxIP}$ as the sole source of Duo1. $GST-Bim1^{185-344}$ immobilized on beads was incubated with soluble cell lysates from these strains, and binding of Duo1-6xFlag was analyzed by Western blot. In agreement with our *in vitro* data, only $Duo1^{WT}$-6xFlag, but not $Duo1^{\Delta SxIP}$-6xFlag, bound to GST-Bim1 (Fig 2D). Deletion of a larger fragment of the C-terminus of Duo1 ($Duo1^{\Delta C}$), which includes the SxIP motif, also prevented binding of Dam1c to GST-Bim1 (Appendix Fig S1C). These data show that formation of the Dam1c-Bim1 complex strictly depends on the EBH domain of Bim1 and the SxIP motif of the Duo1 subunit and that deletion of four amino acids of Dam1c is sufficient for specific disruption of this interaction.

## *In vivo* analysis of the Duo1$^{\Delta SxIP}$ allele reveals reduced metaphase kinetochore localization of Dam1c and a delay in mitotic progression

To study the physiological consequences of a disrupted Dam1c-Bim1 interaction, we constructed yeast strains with 6xFlag-tagged $Duo1^{WT}$ or $Duo1^{\Delta SxIP}$ alleles in a wild-type or Dad1-GFP strain background. The $Duo1^{\Delta SxIP}$ strain grew comparably to a $Duo1^{WT}$ strain on rich media but displayed a mild growth defect at 37°C (Fig 3A). However, C-terminal tagging of Dad1 with GFP in combination with the $Duo1^{\Delta SxIP}$ allele resulted in a severe growth defect at 37°C. FACS analysis of the DNA content of logarithmically growing cells revealed a striking difference between the $Duo1^{WT}$ and the $Duo1^{\Delta SxIP}$ strain. While the culture of the wild-type strain showed an equal fraction of cells with a 1C and 2C DNA content, there was a significant accumulation of cells with a 2C DNA content in case of the $Duo1^{\Delta SxIP}$ strain (Fig 3B) suggesting a delay in mitotic progression. For further analysis of the growth phenotype of the $Duo1^{\Delta SxIP}$ allele, we performed live cell microscopy of $Duo1^{WT}$-6xFlag and $Duo1^{\Delta SxIP}$-6xFlag. For visualization of Dam1c, the Dad1 subunit was C-terminally fused to GFP. In case of the $Duo1^{\Delta SxIP}$ strain, a high number of large budded cells with abnormally large bud size and short inter-kinetochore

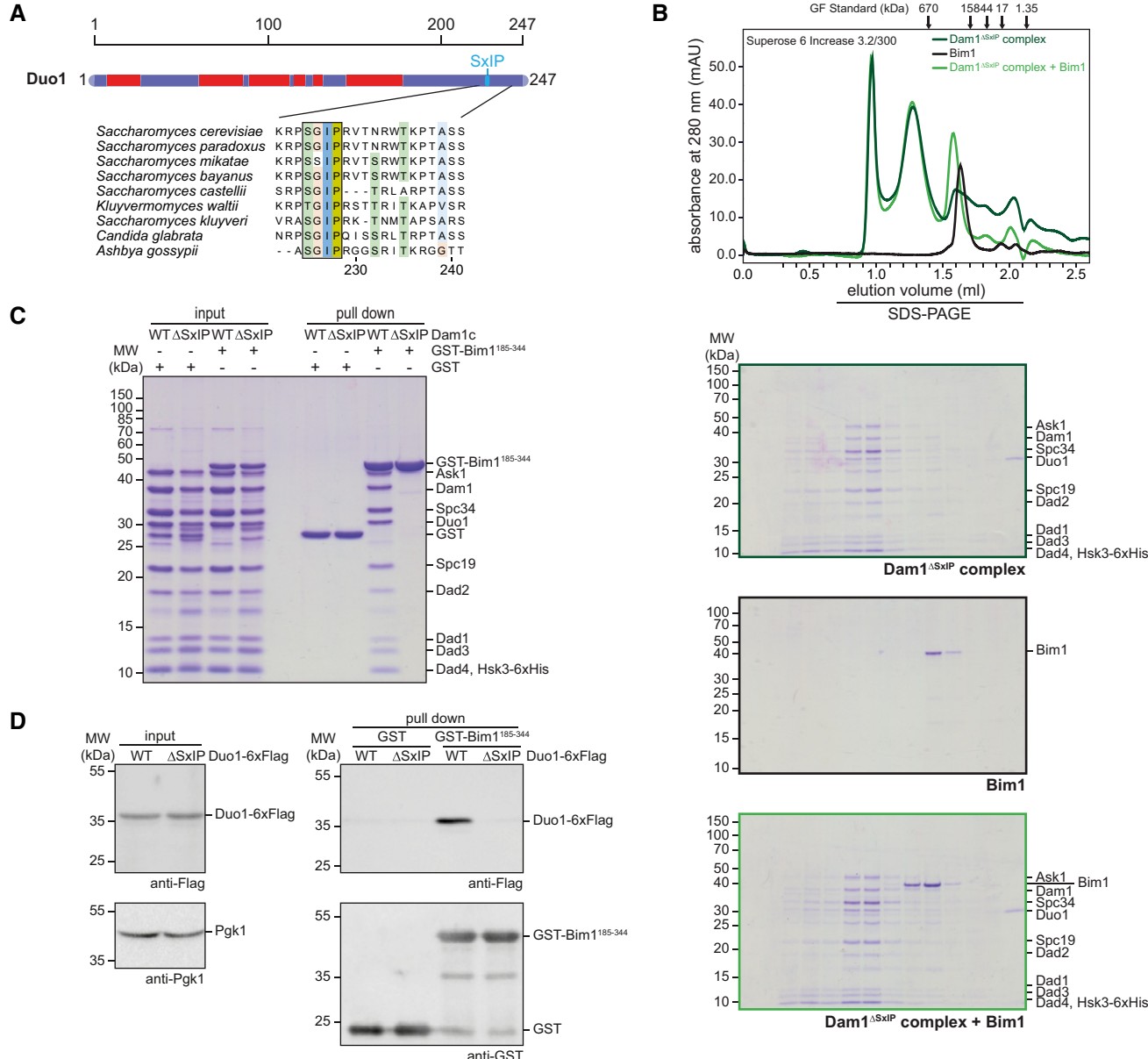

**Figure 2. An SxIP motif located in the C-terminus of Duo1 is required for binding of Bim1 to Dam1c.**

A   Scheme of the Duo1 subunit of the Dam1 complex. Predicted helical regions are highlighted in red, blue regions are unstructured (top). The position of an SxIP motif is marked by a vertical line. The multiple sequence alignment of Duo1 sequences from various fungal species highlights the conservation of the SxIP motif (black box) and adjacent amino acids across species. Amino acids are colored according to the Clustal color scheme.

B   Chromatogram and SDS–PAGE of elution fractions from gel filtration. Binding of Bim1 to Dam1$^{\Delta SxIP}$c lacking the SxIP motif in the C-terminus of Duo1 was examined by analytical size-exclusion chromatography. Elution fractions were analyzed by SDS–PAGE.

C   Solid-phase binding assay to test for binding of different Dam1 complexes to Bim1. Recombinant GST-Bim1$^{185–344}$ was immobilized on glutathione sepharose beads and incubated with either recombinant Dam1$^{WT}$c or Dam1$^{\Delta SxIP}$c. Input and pull-down samples were analyzed by SDS–PAGE.

D   Pull-down assay to compare binding of Dam1$^{WT}$c and Dam1$^{\Delta SxIP}$c with immobilized GST-Bim1$^{185–344}$. Soluble cell lysates from yeast strains expressing the respective Duo1-6xFlag alleles were incubated with GST-Bim1$^{185–344}$ immobilized on beads. Duo1-6xFlag bound to GST-Bim1$^{185–344}$ was analyzed by Western blot. An anti-Pgk1 antibody was used to confirm equal protein amounts in input samples.

distance were observed (Fig 3C and E). The short inter-kinetochore distance clearly indicates these cells as metaphase cells. Time-lapse microscopy revealed that some of these cells stayed in metaphase for 45 min or longer (Movie EV1). The kinetochore clusters showed tumbling motion, suggesting strong spindle movement before the

cells eventually entered anaphase. We did not observe this phenotype in wild-type cells. Notably, both Duo1$^{\Delta SxIP}$ and *bim1Δ* in combination with Dad1-GFP resulted in severe growth defects at 37°C (Appendix Fig S2A). Quantification of Dad1-GFP signal intensities at metaphase kinetochore clusters showed a significant reduction by

20% in the context of the Duo1$^{\Delta SxIP}$ allele compared with Duo1$^{WT}$ (Fig 3D). However, comparable amounts of Dad1-GFP were present at anaphase kinetochore clusters of both Duo1$^{WT}$ and Duo1$^{\Delta SxIP}$ strains. We did not see any difference in cellular Dad1-GFP levels as judged by Western blot (Appendix Fig S2B).

For a more detailed analysis of the cell cycle delay, we analyzed the Pds1/securin degradation kinetics and DNA content of Duo1$^{WT}$ and Duo1$^{\Delta SxIP}$ strains in a Dad1-GFP background. To do so, we arrested the strains with α factor in G1 phase, released, and further cultivated them at 37°C. Samples for Western blot and FACS analysis were taken every 15 min after release. In both strains, Pds1 accumulated until 45 min after release and was subsequently degraded as cells proceeded into anaphase (Fig 3F and G). In the wild-type strain, Pds1 was completely degraded 120 min after release. In contrast, substantial amounts of Pds1 were still detectable in the corresponding sample of the Duo1$^{\Delta SxIP}$ strain. In agreement with these results, the FACS analysis of the cellular DNA content showed an identical cell cycle progression of both strains until 60 min after release. 15 min later, a large number of cells of the wild-type strain have finished mitosis indicated by the reappearance of cells with a 1C DNA content. In contrast, this G1 subpopulation is significantly smaller in the Duo1$^{\Delta SxIP}$ strain at the same time point, allowing the conclusion that the majority of cells has not finished mitosis yet.

Our *in vivo* characterization of the novel Duo1$^{\Delta SxIP}$ allele revealed a delay in mitotic progression, which becomes apparent in live cell microscopy and cell cycle progression analysis. In addition, kinetochore localization of Dad1-GFP was significantly reduced in metaphase compared with a wild-type strain. Thus, we conclude that specific association between Bim1 and Dam1c is required for proper recruitment of Dam1c to metaphase kinetochores.

### The C-terminus of Dam1 acts as an inhibitor for binding of Bim1 to Duo1

A closer look at the lysine–lysine proximity map of the Dam1c-Bim1 complex highlights the large number and high density of crosslinks involving the C-terminus of the Dam1 subunit (Fig 1B). Many lysine residues in this area such as K252, K256, K307, and K320 were crosslinked to the C-terminus of Duo1, which harbors the SxIP motif

(Appendix Table S3). We asked whether removal of the Dam1 C-terminus has any effect on binding of Bim1 to Dam1c. To address this issue, we purified recombinant Dam1–19c from *E. coli*. Dam1–19 is a well-characterized allele of Dam1 with a Q205Stop point mutation that eliminates the C-terminal 139 residues (Cheeseman *et al*, 2001). We found that the C-terminus of Dam1 was not required for binding of Bim1. Moreover, removal of the C-terminus allowed a more complete coelution of Bim1 with Dam1–19c under more stringent binding conditions (400 mM NaCl), while binding to Dam1$^{WT}$c was impaired under these conditions (Fig 4A). We suggest that the Dam1 C-terminus has an inhibitory effect on complex formation *in vitro*. Consistent with this idea, we found that the amount of Dad1-GFP loaded on metaphase kinetochore clusters of a *dam1–19* strain was significantly increased by about 20% compared with a wild-type strain (Fig 4B and C). In contrast, the average anaphase signal intensity was only mildly increased. Cellular Dad1-GFP levels were comparable in wild-type and *dam1–19* strains as judged by Western blot (Appendix Fig 2B). The data reveal that a C-terminal truncation of the Dam1 subunit promotes binding of Bim1 to Dam1c and increases Dad1-GFP localization at metaphase clusters, which is opposite to the effects we observed in Duo1$^{\Delta SxIP}$ strains.

Previous studies showed that the Dam1 subunit is phosphorylated by both Ipl1/Aurora B and Mps1 at multiple residues (Fig 4D) (Cheeseman *et al*, 2002; Shimogawa *et al*, 2006) and truncation of Dam1 in the context of the *dam1–19* allele eliminates the majority of these sites. Based on these observations and the crosslinking results, we propose that the C-terminus of Dam1 acts as an intramolecular inhibitor that partially masks the SxIP motif in the C-terminus of Duo1 (Fig EV2A). By this mechanism, binding of Bim1 to Dam1c is partially inhibited. Removal of the Dam1 C-terminus either by truncation or by phosphorylation may relieve this inhibitory mechanism and promote binding of Bim1 to the complex.

### Overexpression of the Mps1 kinase promotes binding of Bim1 to Dam1c and affects localization of Dad1-GFP in metaphase

We tested the inhibition model by analyzing the effect of specific Mps1 inhibition or overexpression on binding of Bim1 to Dam1c.

---

**Figure 3.  Phenotypic analysis of the Duo1$^{\Delta SxIP}$ allele reveals impaired kinetochore localization of Dam1c and a delay in cell cycle progression.**

A   Serial dilution assay of yeast strains with different Duo1-6xFlag alleles in a wild-type or Dad1-GFP strain background. Growth phenotypes of the respective yeast strains were analyzed by spotting serially diluted amounts of cells on YEPD plates and incubation at different temperatures.

B   FACS analysis of the DNA content of strains with Duo1$^{WT}$-6xFlag or Duo1$^{\Delta SxIP}$-6xFlag alleles with untagged Dad1. Samples were taken from log phase cultures grown at 30°C. Cells with a 1C DNA content are in G1 phase, a 2C DNA content is characteristic of mitotic cells. Cells with an intermediate DNA content are in S phase.

C   Representative images from fluorescence live cell microscopy of Dad1-GFP strains with Duo1$^{WT}$-6xFlag or Duo1$^{\Delta SxIP}$-6xFlag. Displayed cells are in either metaphase (left) or anaphase (right). Scale bar: 2 μm.

D   Quantification of signal intensities of Dad1-GFP at metaphase and anaphase kinetochore clusters of Duo1$^{WT}$-6xFlag and Duo1$^{\Delta SxIP}$-6xFlag strains. Bars represent the mean ± standard deviation. Each spot represents the value measured for a single kinetochore cluster. Values were normalized to the mean value of metaphase clusters of the Duo1$^{WT}$-6xFlag strain. A one-way ANOVA test with Sidak's multiple comparisons test was used to calculate *P*-values. n ≥ 76 kinetochore clusters.

E   Quantification of extra-large (XL) budded cells with short inter-kinetochore distance. Only cells with buds that were grown to more than half of the mother cell's size were counted. Mean ± standard deviation from three independent experiments are displayed. *P*-value was calculated by an unpaired t-test.

F   Analysis of cell cycle progression of yeast strains with different Duo1 alleles. Accumulation and degradation of Pds1 after release from α factor arrest was analyzed by Western blot (left). An antibody against Pgk1 was used to confirm equal protein amounts in samples. The DNA content of the cell population at indicated time points after release from α factor arrest was analyzed by FACS (right). Cells in G1 phase and mitosis are characterized by a 1C and 2C DNA content, respectively. An intermediate DNA content is detected for cells in S phase.

G   Pds1 signal intensities from Western blots shown in F were quantified and corrected for Pgk1 signal of the corresponding sample. Finally, the highest signal intensity (45 min after release) was defined as 1.00 and values for other time points were normalized accordingly.

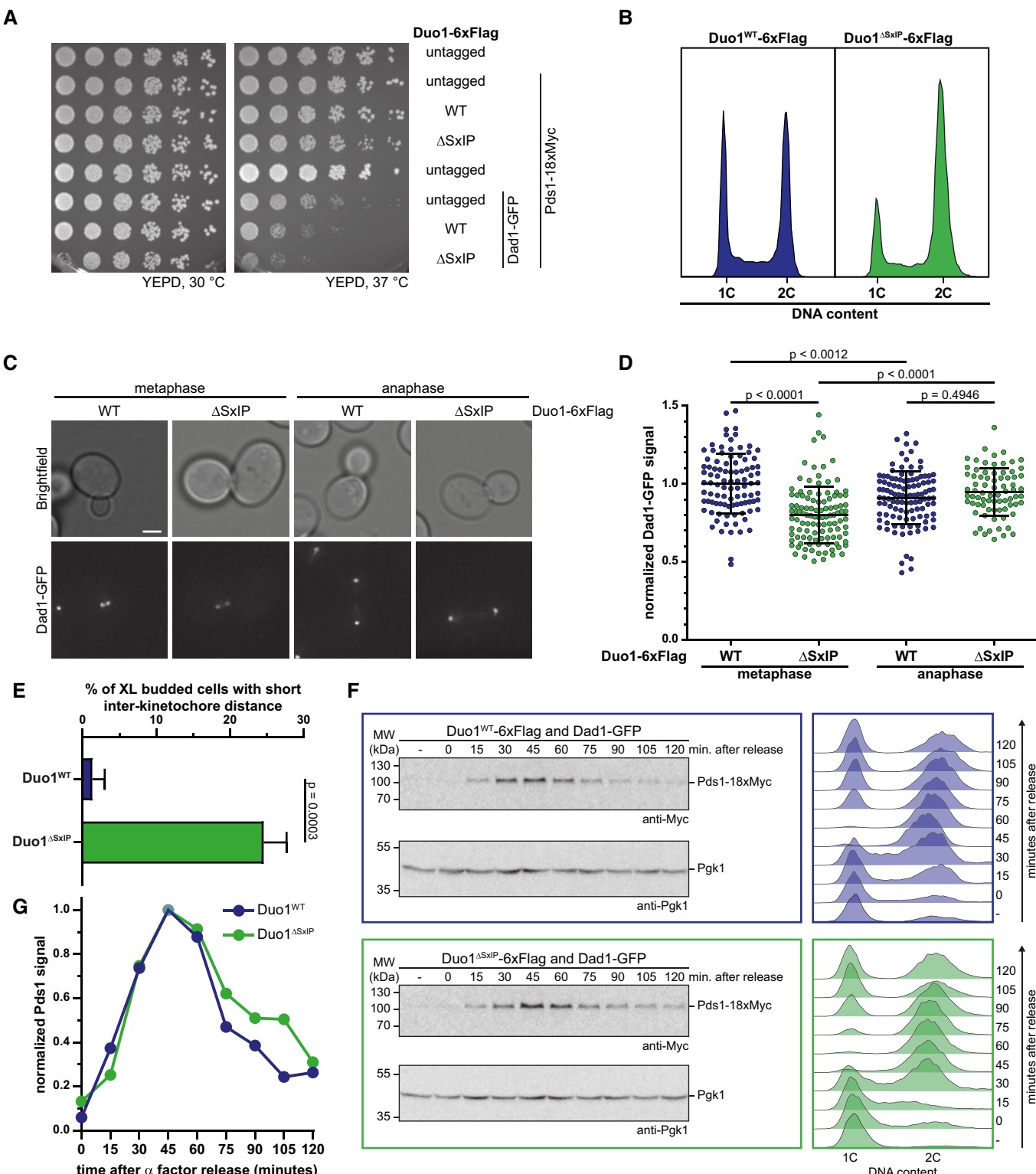

**Figure 3.**

First, we tested binding of Bim1 using cell lysates from a strain with the temperature-sensitive *mps1–737* allele (Schutz & Winey, 1998). This allows for the selective inhibition of Mps1 by growing the cells at the restrictive temperature of 37°C. Binding of Duo1-6xFlag to immobilized GST-Bim1[185–344] was reduced when the *mps1–737* strain was grown at the restrictive temperature of 37°C (Fig 4E). In contrast, binding was not compromised when the cells were grown at the permissive temperature of 30°C. In an Mps1[WT] strain, the same amounts of Duo1 bound to Bim1 irrespective of the temperature at which the cells were grown. We observed a similar effect

when inhibiting Mps1-as1 by the ATP analog 1NM-PP1 (Jones *et al*, 2005, Appendix Fig S3). In a complementary approach, we constructed a yeast strain in which an additional copy of 1xMyc-Mps1 is integrated under control of the inducible Gal promoter (pGal-Mps1). In the absence of glucose and presence of galactose, Mps1 is overexpressed in this strain. We prepared soluble cell extracts from Mps1[WT] and pGal-Mps1 strains grown in the presence of galactose and tested binding of Duo1-6xFlag to immobilized GST-Bim1[185–344]. Overexpression of Mps1 from a galactose-inducible promoter resulted in a significant increase in binding of Dam1c to Bim1 compared with the non-overexpression strain (Fig 4F). Taken together, our results show that Mps1 kinase activity enhances binding of Bim1 to Dam1c and is required for efficient complex formation.

To analyze the effect of Mps1 overexpression on the localization of Dam1c, we performed live cell microscopy in a Dad1-GFP strain background. In addition, Ndc80 was fused to mRuby2 (Ndc80-mRuby2) as reference point for the localization of kinetochores. As described before, cells overexpressing Mps1 arrested in metaphase (Hardwick *et al*, 1996). In about 77% of metaphase cells of an Mps1[WT] strain, the two kinetochore clusters were clearly separated (bi-lobed) as judged by both Dad1-GFP and Ndc80-mRuby2 localization (Fig 4G–I). In contrast, Dad1-GFP had two distinct appearances upon overexpression of Mps1. Only 28% of metaphase cells had bi-lobed Dad1-GFP localization as found in Mps1[WT] cells. The majority of metaphase cells (72%) displayed a bar-shaped Dad1-GFP localization with no clear separation of the two clusters (Fig 4G, middle panel, and Fig 4H and I). In contrast to Dad1-GFP, Ndc80-mRuby2 still localized as two distinct foci, indicating that overexpression of Mps1 mainly affects localization of Dad1-GFP and does not lead to a complete dis-organization of kinetochores. The altered localization of Dad1-GFP compared with Ndc80-mRuby2 is furthermore highlighted by line scans along the axis between the kinetochore clusters

(Fig 4H). For Mps1[WT] cells, the profile showed two clearly separated peaks for both Dad1-GFP and Ndc80-mRuby2. In contrast, the two Dad1-GFP peaks were less clearly separated in pGal-Mps1 cells and displayed increased signal intensity. To address whether Dad1-GFP localization is indeed affected by overexpression of Mps1 and not simply a consequence of a metaphase arrest, we arrested Dad1-GFP cells in metaphase by depletion of Cdc20 using an auxin-inducible degradation system (Cdc20-AID-9xMyc; Nishimura *et al*, 2009). Cdc20 was depleted after addition of the auxin analog naphthalene acetic acid (NAA) and cells arrested in metaphase (Fig EV2B and C). Only 39% of cells arrested in metaphase exhibited a bar-shaped localization of Dad1-GFP, which is substantially less than in Mps1-overexpressing cells (Figs 4I and EV2D). Thus, we conclude that Mps1 kinase activity and not merely the metaphase arrest affects Dad1-GFP localization.

In addition to the bar-shaped Dad1-GFP clusters, Mps1-overexpressing cells frequently showed elongated structures emerging from or close by the kinetochore clusters (Fig 4G, bottom panel). These structures were decorated by Dad1-GFP, but not Ndc80-mRuby2. Assuming that Dam1c exclusively localizes to kinetochores and microtubules, we speculate that these structures are random nuclear microtubules. Localization in the nucleoplasm was confirmed by live cell microscopy of Mps1-overexpressing strain in which the nuclear membrane was labeled by Nup60-RedStar2 (Fig EV2E).

Furthermore, we acquired evidence that the Dam1 subunit may not be the only relevant substrate for Mps1 in this process. In addition to the previously described substrate Dam1 (Shimogawa *et al*, 2006), recombinant Mps1 also phosphorylated the Ask1 and Duo1 subunits, as well as Bim1 itself, to a similar extent *in vitro* (Fig EV3A). We thus tested the effect of Mps1 phosphorylation on Bim1 binding *in vitro* by size-exclusion chromatography under high salt conditions (400 mM NaCl). To this end, we first incubated Dam1c with Mps1 in the absence or presence of ATP. SDS–PAGE

**Figure 4. Mps1-dependent modification of the Dam1 C-terminus regulates the interaction with Bim1.**

A    SDS–PAGE analysis of elution fractions from analytical size-exclusion chromatography. Binding of Bim1 to Dam1[WT]c and Dam1–19c was analyzed by size-exclusion chromatography. The gel filtration buffer contained 400 mM NaCl.

B    Representative images from live cell microscopy of Dam1[WT] and *dam1–19* strains with Dad1-GFP. Displayed cells are in metaphase (left) or anaphase (right). Scale bar: 2 μm.

C    Quantification of Dad1-GFP signal intensities at metaphase and anaphase kinetochore clusters of Dam1[WT] and *dam1–19* strains. Bars represent the mean ± standard deviation. Each spot represents the value measured for a single kinetochore cluster. Values were normalized to the mean value of metaphase clusters of the Dam1[WT] strain. *P*-values were obtained by a one-way ANOVA test with Sidak's multiple comparisons test. $n \geq 86$ kinetochore clusters.

D    Schematic representation of the Dam1 protein. Previously characterized Ipl1 and Mps1 phosphorylation sites are annotated. Dam1–19 is truncated by a Q205Stop point mutation. The C-terminus of Dam1 including several phosphorylation sites framed by the dashed box is missing in *dam1–19*.

E, F    GST-Bim1[185–344] pull-down assays. Binding of immobilized GST-Bim1[185–344] to Duo1-6xFlag was analyzed in context of (E) selective inhibition of Mps1 (*mps1-737*) or overexpression of Mps1 from a galactose-inducible promoter (F). For inhibition of Mps1, cells were grown at the restrictive temperature of 37°C. Overexpression of Mps1 was achieved by growing cells in medium containing galactose. GST-Bim1[185–344] immobilized on glutathione sepharose beads was incubated with soluble cell lysates of the respective strains grown under the mentioned conditions. Binding of Duo1 was analyzed by Western blot. Pgk1 indicates equal protein amounts in all samples.

G    Live cell microscopy of a Dad1-GFP strain under conditions of Mps1 overexpression. The localization of Dam1c was visualized by Dad1-GFP in either an Mps1[WT] or an Mps1 overexpression setup (pGal-Mps1). Ndc80-mRuby2 was used as reference point for the localization of kinetochores. Images show representative metaphase cells from both strains. The area in the white dashed box is shown as magnification in the lower left corner. White arrowheads point at nuclear microtubules decorated by Dad1-GFP. Please note that brightness and contrast of the Dad1-GFP image in the lower row were optimized to make the nuclear microtubules clearly visible. Scale bar: 2 μm.

H    Representative line scans of Dad1-GFP and Ndc80-mRuby2 signal intensities along the mitotic spindle. Displayed data were acquired from the cells shown in G (labeled with white dashed boxes).

I    Quantification of Dad1-GFP cluster shape in Mps1[WT] and pGal-Mps1 strains. Bi-lobed cluster shape with clearly separated Dad1-GFP foci was distinguished from bar-shaped cluster without clear separation. The mean ± standard deviation from three independent experiments are shown. *P*-values were calculated with a two-way ANOVA test with Sidak's test for multiple comparisons.

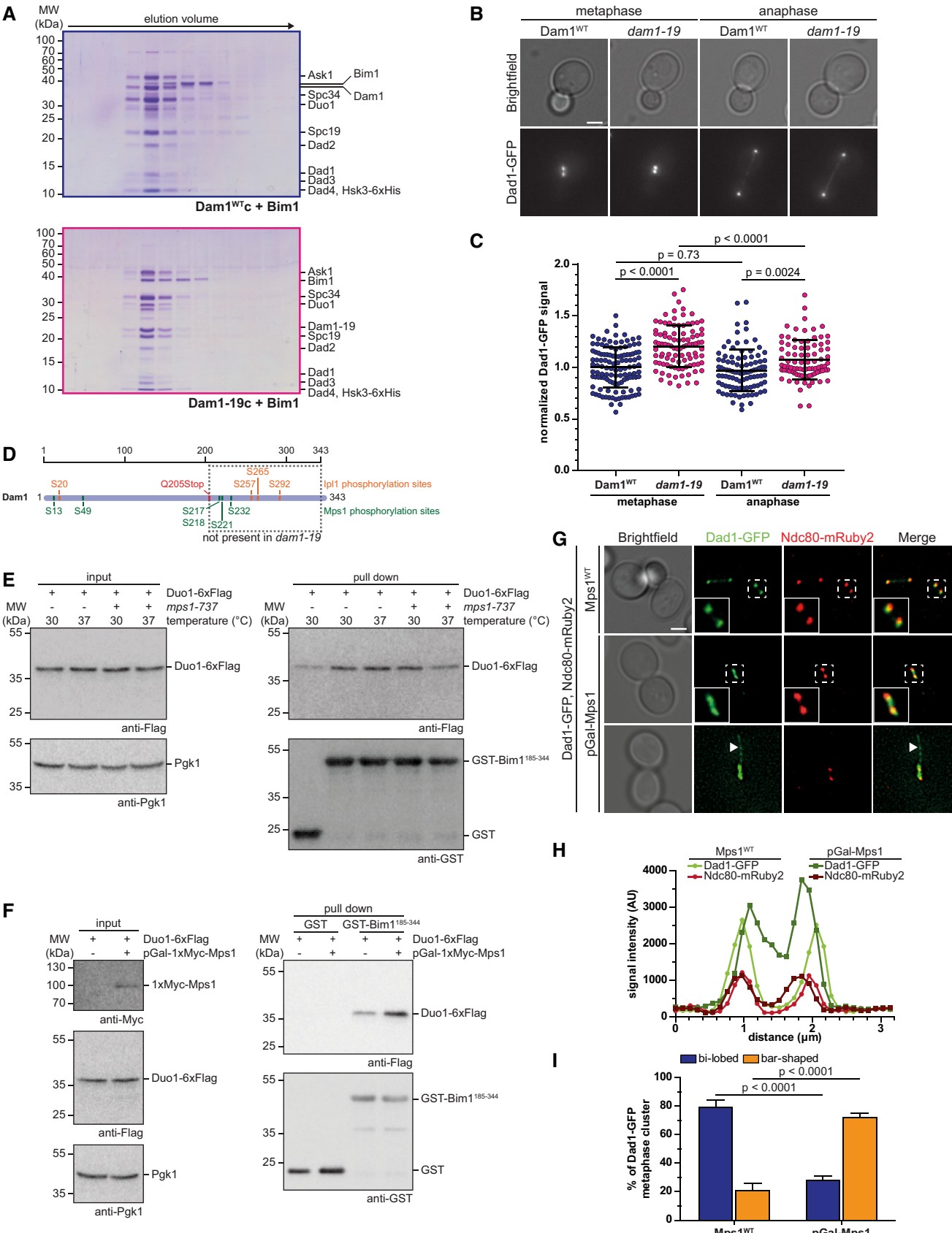

**Figure 4.**

analysis of phosphorylated Dam1c or Bim1 revealed slower migrating forms of Ask1, Dam1, Duo1, and Bim1 compared with unphosphorylated proteins, suggesting efficient *in vitro* phosphorylation. Phosphorylated Dam1c eluted later during gel filtration compared with the unphosphorylated complex (Fig EV3C and D), indicative of a shift to smaller oligomeric or monomeric complex species. The majority of Bim1 was eluted independently from unphosphorylated Dam1c under the chosen salt concentration. However, we found an increased proportion of Bim1 coeluting with Dam1c after phosphorylation by Mps1 (Fig EV3C and D), which resembles our observation of enhanced Bim1 binding to the Dam1–19 complex (Fig 4A).

Taken together, we conclude that Mps1 phosphorylation has two distinct effects *in vitro*: It apparently reduces oligomerization of Dam1c, an effect that is partially redundant with phosphorylation by Ipl1, while simultaneously promoting binding of Bim1 to Dam1c consistent with the effect of Mps1 overexpression in cells.

### Bim1 recruits Bik1 to Dam1c and promotes its oligomerization in solution

In addition to Bim1, microtubule-associated proteins such as Stu2 and Bik1 are present at dynamic microtubule plus ends (Akhmanova & Steinmetz, 2008, 2015). As shown in Fig EV1A, Stu2 does not bind to Dam1c. We tested binding of Bik1 to Dam1c using recombinant Bik1 purified from *E. coli*. Gel filtration of Bik1 together with Dam1c revealed that the majority of Bik1 eluted independently from Dam1c (Fig EV4A). However, in the presence of Bik1 a small fraction of Dam1c eluted quite early (at about 0.9–1.0 ml), suggesting formation of large assemblies of Dam1c. Western blot analysis of the elution fractions showed only a small fraction of Bik1 coeluting with Dam1c under these conditions (Fig EV4A). Adding Bim1 to the binding reaction resulted in increased Bim1-dependent recruitment of Bik1 to Dam1c and enhanced formation of large Dam1c oligomers (Figs 5A and B, and EV4A). Increasing the input concentrations of all three components (Dam1c from 5 to 7.5 μM and 10 μM) gradually promoted the formation of high molecular weight Dam1c species (Fig 5A and B). The retention volume of Dam1c itself depended on the input concentration of the complex with high protein concentrations leading to earlier elution in size-exclusion chromatography (Fig EV4B). As described before, Bim1 and Bik1 formed a stable complex in solution (Fig 5A and B, Blake-Hodek *et al*, 2010).

If the formation of large Dam1c assemblies is a consequence of Bim1-Bik1-induced oligomerization, we would not expect to see this effect when using an oligomerization-deficient Dam1 complex. To test this hypothesis, we used recombinant Dam1$^{4D}$c, which mimics Ipl1-dependent phosphorylation of the Dam1 subunit at four sites (S20, S257, S265, and S292) and is deficient in forming oligomers (Cheeseman *et al*, 2002; Wang *et al*, 2007; Zelter *et al*, 2015). Indeed, the Bim1-Bik1 complex bound to Dam1$^{4D}$c in solution, but we did not observe the formation of large Dam1c assemblies (Fig EV4C). Thus, we conclude that Bim1 recruits Bik1 to Dam1c, which induces the oligomerization of the complex in solution. The experiment also demonstrates that Ipl1 phosphorylation of Dam1 does not inhibit binding to Bim1-Bik1, but does prevent the Bim1-Bik1-induced oligomerization of Dam1c.

Furthermore, we found that using increased input concentrations of Dam1c, Bim1, and Bik1 in combination with lowering the ionic strength of the gel filtration buffer (100 mM NaCl) led to very early elution of the complex, indicating a shift from small toward larger oligomers (Fig 5C). Samples of the early eluting Dam1c-Bim1-Bik1 complex were analyzed by electron microscopy. Strikingly, these samples revealed homogenous, fully formed rings as the most prominent species (Fig 5D). In contrast, ring-like structures presented only a minor fraction of the Dam1c-Bim1 sample presented in Fig 1F. The Dam1c-Bim1-Bik1 rings are remarkably stable, as they could be separated by gel filtration chromatography from smaller particles that were probably still present because of deviations from an ideal stoichiometry, impurities, and/or partially unfolded proteins (Fig 5E). The Dam1c-Bim1-Bik1 rings show no sign of dissociation with smaller species at the chosen low salt conditions. They are characterized by an outer diameter of 60 nm and are thicker than rings assembled in the presence of Bim1 alone, giving them a wreath-like appearance. Additional mass visible in the center of the rings might be due to flexible domains derived from Bim1 or Bik1 that have been flattened on the EM grid. The smaller oligomers seen in the micrographs are in full agreement with the different states of Dam1c oligomerization found via Western blot of Dam1c alone, Dam1c in the presence of Bim1, and Dam1c in the presence of both Bim1 and Bik1 (Figs 1E and F, 5D and E, and EV4A). These experiments provide the first example that Dam1c can assemble into full rings in solution in the absence of microtubules. Previous studies only reported oligomerization of Dam1c on surfaces such as microtubules or negatively charged phospholipids (Miranda *et al*, 2005; Westermann *et al*, 2005), or employed oligomerization of engineered constructs into helical tubes (Jenni & Harrison, 2018), which is unlikely to represent a physiological assembly mechanism.

**Figure 5.  Bim1 recruits Bik1 to Dam1c and induces oligomerization of the complex in a concentration-dependent manner.**

A, B  5, 7.5, or 10 μM Dam1$^{WT}$c alone or in combination with respective concentrations of Bim1-Bik1 complex was analyzed by size-exclusion chromatography. SDS–PAGE analysis of elution fractions is shown in (B). High molecular weight species of Dam1$^{WT}$c were eluted after 0.9–1.0 ml (see chromatogram in (A)). SDS–PAGE analysis of Dam1c in the absence of Bim1-Bik1 is shown in Fig EV4B.

C  Chromatogram of size-exclusion chromatography of Dam1c with Bim1 and Bik1 and SDS–PAGE analysis of elution fractions. The elution buffer contained 100 mM NaCl in contrast to 200 mM in previous experiments. Please note that under these conditions, the equilibrium is shifted toward the high oligomeric complex. Dashed boxes indicate elution fractions used for EM analysis shown in D.

D, E  EM images of negatively stained Dam1c in the presence of Bim1 and Bik1 at low salt conditions (10 mM NaCl). The final concentrations of Dam1c, Bim1, and Bik1 were 0.3, 0.1, and 0.25 μM, respectively. The Dam1c-Bim1-Bik1 complex could be further purified by gel filtration in presence of 100 mM NaCl (20 mM HEPES, pH 7.4, 100 mM NaCl, 1 mM MgCl$_2$, and 0.1 mM TCEP) prior to the final dilution to low salt conditions for EM (D). The scale bars are 100 nm, boxes mark representative complexes shown as magnified inserts.

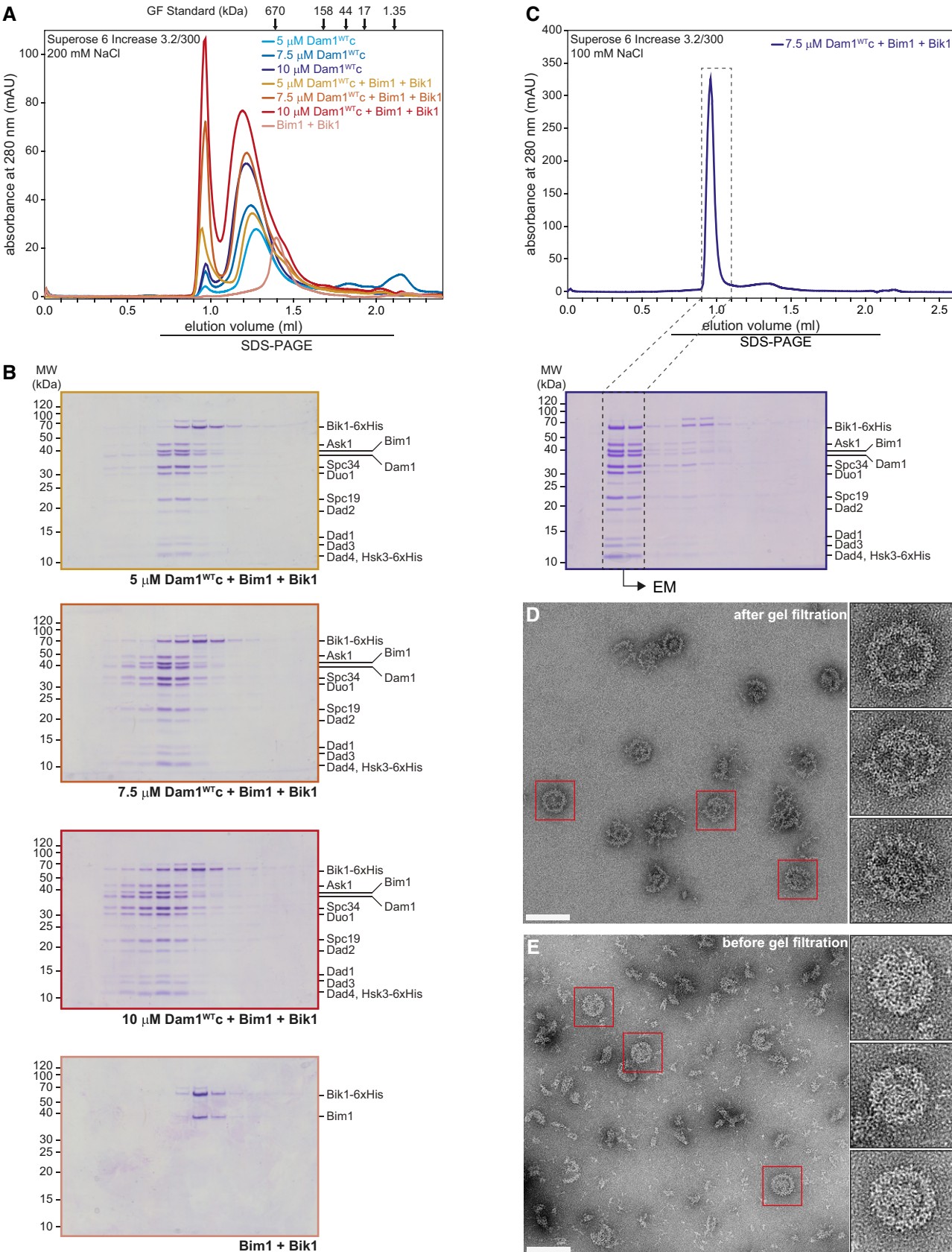

**Figure 5.**

## Bim1 and Ndc80c may represent alternative binding partners of Dam1c at the kinetochore

The interaction between Dam1c and Ndc80c is a key step for establishing stable and tension-bearing end-on attachments (Lampert et al, 2010; Tien et al, 2010). Previous studies biochemically analyzed binding of Dam1c and Ndc80c (Lampert et al, 2010, 2013; Tien et al, 2010). However, at this point it is unclear whether binding of Dam1c to Ndc80c is compatible with simultaneous binding of Bim1 to Dam1c or whether Bim1 is removed upon binding of Ndc80c to Dam1c (Fig 6A). To address this question, we performed a pull-down assay with recombinant Flag-tagged Ndc80c immobilized on beads. Either Dam1c or Bim1 alone or Dam1c with increasing amounts of Bim1 was added to the loaded beads. Ndc80c-Flag and bound proteins were eluted from the beads with 3xFlag peptide, and samples were analyzed by SDS–PAGE (Fig 6B). Using this solid-phase binding assay, we found that Dam1c bound robustly to insect cell-expressed Ndc80c (lane 3), while Bim1 only very weakly associated with Ndc80c (lane 4). Since Bim1 did not bind to Ndc80c in size-exclusion chromatography (Fig EV5A), we speculated that Bim1 spuriously binds to the beads loaded with Ndc80c, which is additionally favored by the relatively high Bim1 concentrations (up to 16 μM). Increasing the Bim1 input concentration up to 16 μM gradually decreased the amount of Dam1c bound to Ndc80c by about 30% (Fig 6B, lanes 5–9, Fig 6C). We conclude that binding of Bim1 to Dam1c inhibits the interaction with Ndc80c under these conditions.

To further address this point, we performed size-exclusion chromatography combining Dam1c, Bim1, Bik1, and Ndc80c (Fig EV5B and C). Dam1c and Ndc80c formed a stable complex in solution that eluted earlier from the column than the individual complexes. In addition, we found a small proportion of Dam1c-Ndc80c complex eluting close to the void at a similar position as the Dam1c-Bim1-Bik1 complex. To test binding of Ndc80c to larger Dam1c assemblies, we pre-assembled Dam1c in the presence of Bim1 and Bik1 as described before and added Ndc80c. Under these conditions, the majority of Ndc80c eluted at a position similar to Dam1c-Ndc80c complex, while only a minor amount of Ndc80c coeluted with the largest Dam1c-Bim1-Bik1 species. Thus, rather than observing an increased formation of high molecular weight assemblies containing all components, we interpret these data as additional evidence for distinct assemblies consisting of Dam1c-Bim1-Bik1 or Dam1c-Ndc80c.

Based on the result of our in vitro approach, we speculate that binding of Dam1c to Bim1 and to Ndc80c represent two consecutive steps during the process of attachment formation. We analyzed the relevance of both interactions for cell growth in vivo by combining the Duo1$^{\Delta SxIP}$ allele with the Ndc80$^{\Delta 490-510}$ allele, which lacks a part of a conserved loop region. Yeast strains carrying this allele show severe growth defects and are impaired in formation of end-on attachments. Furthermore, binding of Ndc80c to Dam1c and localization of Dam1c to kinetochores were impaired in this mutant (Maure et al, 2011). Since the Ndc80$^{\Delta 490-510}$ allele itself causes severe growth defects, we used a conditional FRB system for our genetic analysis. Endogenous Ndc80 was fused to FRB-GFP, which allows for efficient removal of the FRB-tagged Ndc80c from the nucleus in the presence of rapamycin. In addition, Ndc80$^{WT}$ or Ndc80$^{\Delta 490-510}$ was integrated at the LEU2 locus and expressed under

control of the Ndc80 promoter. Growth of yeast strains with either Duo1$^{WT}$ or Duo1$^{\Delta SxIP}$ in combination with Ndc80$^{WT}$ or Ndc80$^{\Delta 490-510}$ was analyzed in a serial dilution assay. Cells were grown at different temperatures in the absence or presence of rapamycin (Fig 6D). In the absence of rapamycin, cells carrying the Ndc80$^{\Delta 490-510}$ allele and Duo1$^{WT}$ displayed reduced growth after incubation at 37°C. In contrast, the combination of Duo1$^{\Delta SxIP}$ and Ndc80$^{\Delta 490-510}$ allowed no growth under the same condition. We assume that Ndc80$^{\Delta 490-510}$ has a negative dominant effect over the FRB-GFP-tagged Ndc80$^{WT}$, which is then further exacerbated by Duo1$^{\Delta SxIP}$. In the presence of rapamycin, the FRB-GFP-tagged Ndc80$^{WT}$ is removed from the nucleus. Ndc80$^{WT}$ integrated at the LEU2 locus supports cell growth at all tested incubation temperatures irrespective of the Duo1 allele (Fig 6D, lower panel). The Ndc80$^{\Delta 490-510}$ allele supported growth at 25°C, while cell growth was completely impaired at 37°C. In addition, a strain carrying the Duo1$^{WT}$ and Ndc80$^{\Delta 490-510}$ was severely compromised in growth when incubated at 30°C. The Duo1$^{\Delta SxIP}$ allele partially suppressed the growth defect caused by the Ndc80$^{\Delta 490-510}$ allele at this temperature. In line with the idea of consecutive binding steps, we speculate that disturbed binding of Ndc80$^{\Delta 490-510}$ to Dam1c is lethal under these conditions. However, the Duo1$^{\Delta SxIP}$ allele prevents binding of Bim1 to Dam1c, which competes with Ndc80c for binding to Dam1c and thus enables the interaction between Ndc80c and Dam1c, alleviating the growth defect under these conditions.

## Disruption of multiple Dam1c oligomerization pathways reduces cell growth in vivo

Previous studies identified different factors, which regulate oligomerization of the Dam1 complex. Phosphorylation of Dam1 S20 by Ipl1 prevents oligomerization of the complex, while phosphorylation of Ask1 S216 and S250 by Cdk1 promotes oligomerization (Fig 7A; Wang et al, 2007; Zelter et al, 2015; Gutierrez et al, 2020). Here, we show that binding of Bim1 and Bik1 induces oligomerization of Dam1c (Fig 5). We tested how cell growth is affected if two of these pathways are disrupted. To do so, we generated yeast strains, which combine either the Duo1$^{WT}$ or Duo1$^{\Delta SxIP}$ allele with Ask1$^{WT}$, Ask1$^{2A}$, or Ask1$^{2D}$, which prevents or mimics Cdk1-dependent phosphorylation of Ask1 S216 and S250, respectively. Ask1$^{2A}$ and Ask1$^{2D}$ did not affect cell growth in a Duo1$^{WT}$ strain background (Fig 7B). In contrast, Ask1$^{2A}$ compromised growth of yeast strains carrying the Duo1$^{\Delta SxIP}$ allele, particularly when cells were grown at low temperature such as 25 or 16°C. The Ask1$^{2D}$ allele did not reduce growth of Duo1$^{\Delta SxIP}$ strains. Our results show that disruption of individual Dam1c oligomerization mechanisms does not affect cell growth. However, interference with both Bim1-dependent and Ask1 phosphorylation-dependent oligomerization severely reduces cell growth, highlighting the importance of Dam1c oligomerization for faithful chromosome segregation.

## Discussion

In this study, we reveal how general plus end-associated proteins are specifically re-purposed to control outer kinetochore assembly. We report and characterize the interaction between the Dam1 complex (Dam1c) and the autonomous plus end-tracking protein

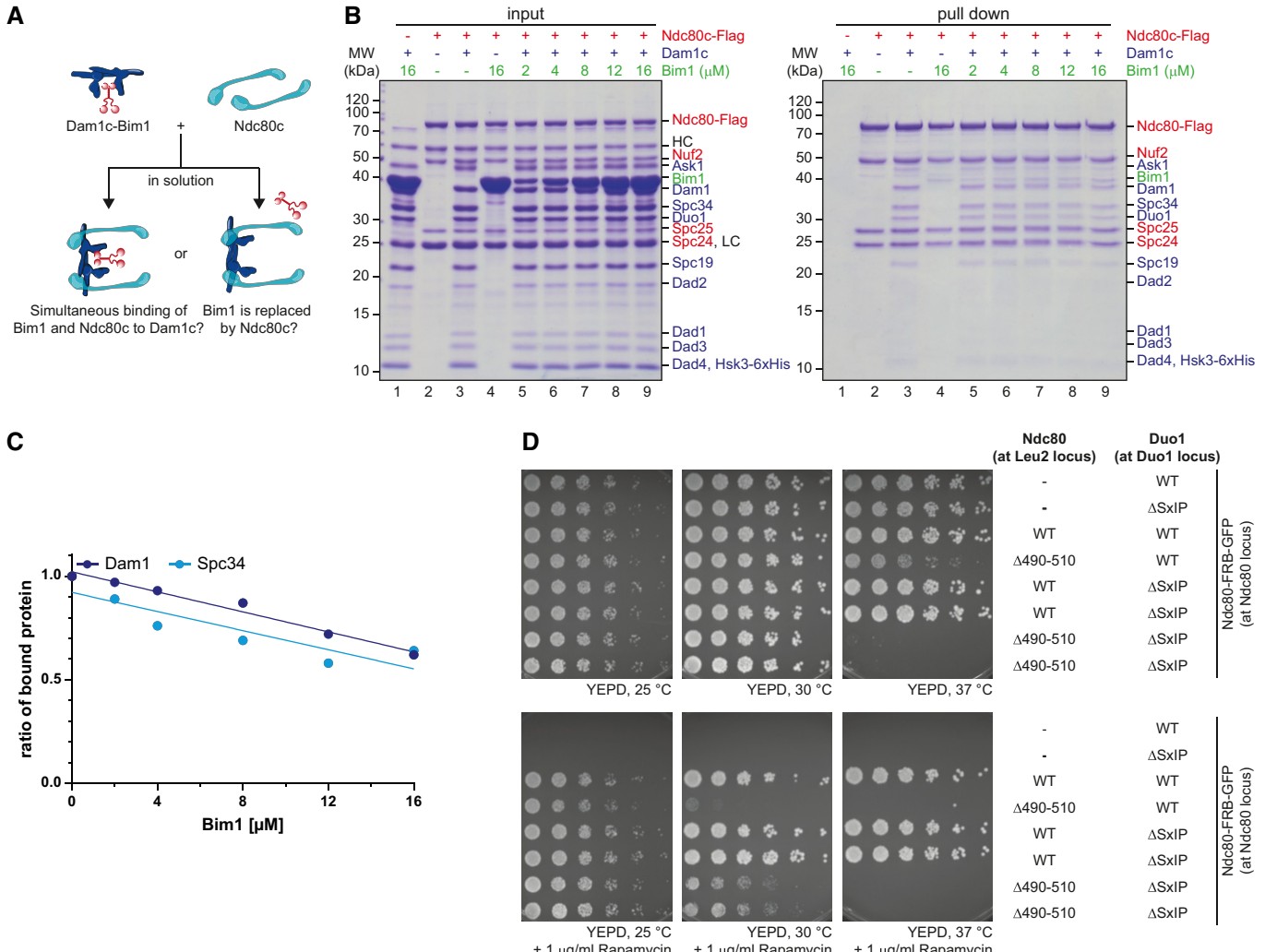

**Figure 6. Formation of Dam1c-Bim1 complex and formation of Dam1c-Ndc80c complex are two consecutive steps during establishment of end-on attachments.**

A  Illustration of possible complex formation involving Dam1c, Bim1, and Ndc80c. After addition of Ndc80c to pre-formed Dam1c-Bim1 complex in solution, two possible outcomes are expected. Either Dam1c and Ndc80c form a complex including Bim1 or Bim1 is replaced upon binding of Ndc80c.

B  Pull-down assay for analysis of Dam1c-Bim1 binding to Ndc80c. Ndc80c immobilized on beads was incubated with either Dam1c or Bim1 alone or with pre-formed Dam1c-Bim1 complex with increasing amounts of Bim1. Input and elution samples were analyzed by SDS–PAGE. LC and HC denote the light and heavy chains of the M2 anti-Flag antibody used for immobilization of Ndc80c on beads. Protein names in blue, red, and green refer to Dam1c, Ndc80c, and Bim1, respectively.

C  Quantification of Dam1 and Spc34 signal intensities in pull-down samples shown in (B). Signal intensities from samples without Bim1 were normalized to the value of 1.0. Linear regression was calculated based on the data points.

D  Serial dilution assay of yeast strains with different Duo1 and Ndc80 alleles in an Ndc80-FRB-GFP background. Ndc80[WT] or Ndc80[Δ490–510] integration constructs were integrated at the LEU2 locus and combined with either the Duo1[WT] or Duo1[ΔSxIP] allele (at DUO1 locus). Cells were spotted on YEPD medium without and with 1 μg/ml rapamycin and incubated at the indicated temperatures.

Bim1. Deletion of a conserved SxIP motif in the C-terminus of the Duo1 subunit of Dam1c was sufficient to disrupt this interaction. Our *in vivo* analysis revealed a significant reduction in kinetochore localization of Dam1c during metaphase in combination with a delay in mitotic progression. Bim1 additionally recruits Bik1 to Dam1c and triggers oligomerization into Dam1c rings. Furthermore, we identified Mps1 as a regulator of this interaction and identified additional Dam1c subunits and Bim1 as novel Mps1 substrates. EM analysis of the Dam1c-Bim1 complex revealed that Bim1 localizes in

close proximity to the protrusion domain. This configuration of the Dam1c-Bim1 complex might be important for correct positioning of Dam1c relative to the kinetochore and will be discussed later.

## Molecular mechanisms of Dam1c recruitment to the yeast kinetochore

Members of the family of EB proteins such as Bim1 bind their cargos by two distinct mechanisms: by binding either to SxIP motifs via

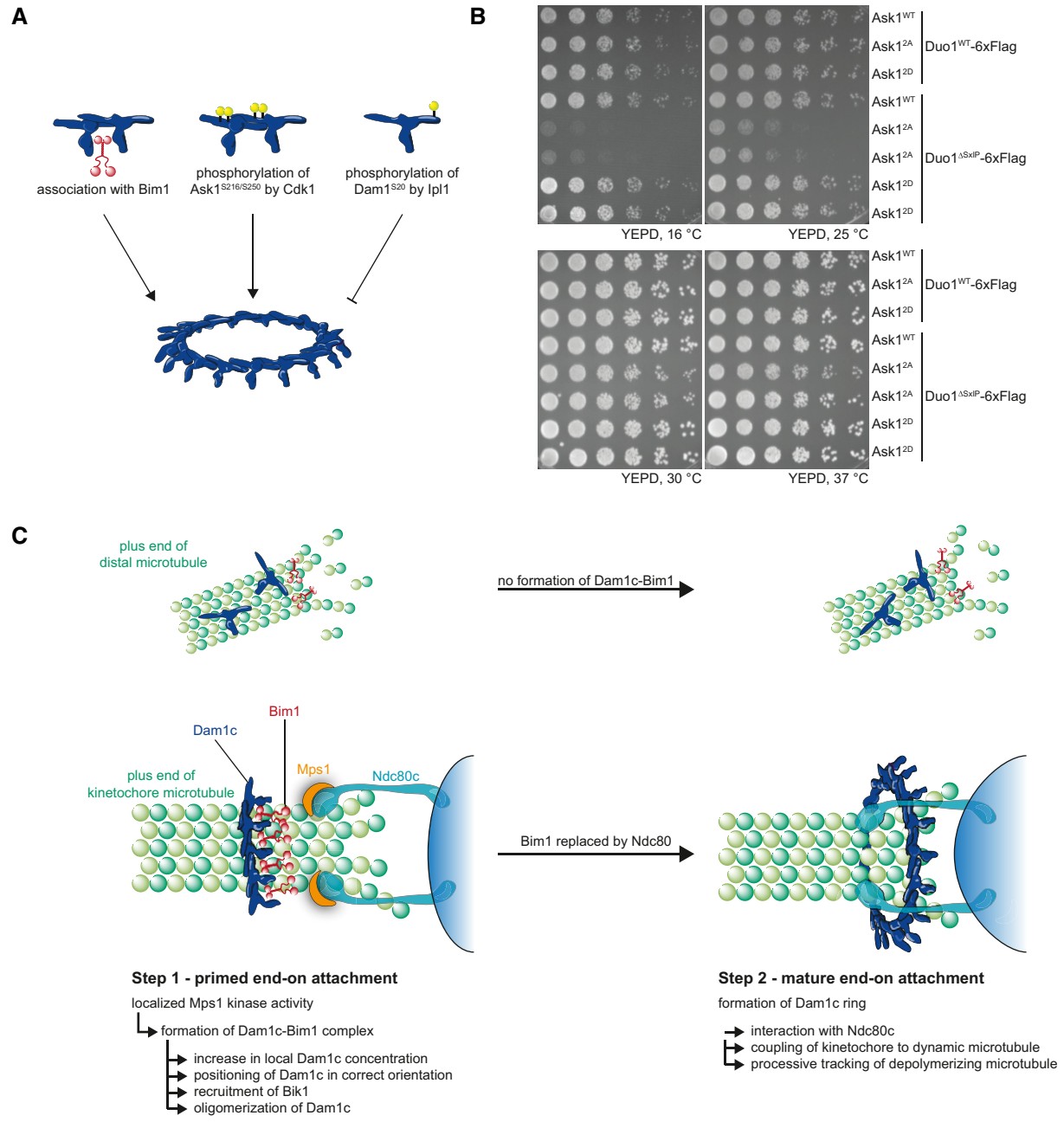

**Figure 7. Duo1$^{\Delta SxIP}$ and Ask1$^{2A}$ alleles cause a synthetic growth defect.**

A  Model of regulatory mechanisms of Dam1c oligomerization. Both binding of Bim1 to Dam1c and Cdk1-dependent phosphorylation of Ask1 promote oligomerization of the complex, while phosphorylation of Dam1 by Ipl1 prevents oligomerization.
B  Serial dilution assay of yeast strains with 6xFlag-tagged Duo1$^{WT}$ or Duo1$^{\Delta SxIP}$ and Ask1$^{WT}$, Ask1$^{2A}$, or Ask1$^{2D}$. Cells were grown at the indicated temperatures, and images were taken 2 days (25, 30, and 37°C) or 4 days (16°C) after spotting.
C  Mps1 localizes to unattached kinetochores and phosphorylates Dam1c on proximal microtubules (left). By this, Bim1 associates with Dam1c and the complex is enriched at microtubule plus ends, which allows formation of end-on attachments (right). Since phosphorylation of Dam1c does not happen on distal microtubules, association with Bim1 and accumulation at plus ends are restricted to microtubules, which are close to unattached kinetochores.

their EBH domain or to CAP-Gly domains via their C-terminal EEF/ Y motif (ETF in Bim1, Honnappa *et al*, 2009; Weisbrich *et al*, 2007). Our *in vitro* reconstitution of the Dam1c-Bim1 complex clearly shows that this interaction depends on the SxIP motif of Duo1. Binding of Bim1 to Dam1c *in vivo* might be limited by two factors: First,

affinities of EBH domains for SxIP motif-containing peptides were described to be rather low (Honnappa *et al*, 2005, 2009). Second, Dam1c presumably competes with a large number of other proteins for binding to Bim1. Increasing the number of Bim1 binding sites by forming Dam1c homodimers possibly increases binding of Bim1.

Furthermore, our results show that Mps1-dependent phosphorylation of Dam1c increases the affinity of Dam1c for Bim1 if required.

Disruption of the Dam1c-Bim1 complex *in vivo* using the Duo1$^{\Delta SxIP}$ complex has two effects. First, the amount of Dam1c localized to metaphase kinetochore clusters is significantly reduced as judged by quantification of Dad1-GFP intensities (Fig 3D). Second, cells with the Duo1$^{\Delta SxIP}$ allele are delayed in cell cycle progression, which is apparent in live cell microscopy and analysis of Pds1 degradation (Movie EV1 and Fig 3F and G). We conclude that Bim1 is required for loading of sufficient amounts of Dam1c to kinetochores to allow faithful chromosome segregation. The relatively mild phenotype of the Duo1$^{\Delta SxIP}$ allele alone is explained by the parallel Cdk1-dependent pathway that promotes Dam1c oligomerization. In addition, we have previously shown that Dam1c displays some preference for microtubule plus ends, even in the absence of Bim1 (Lampert *et al*, 2010). Previous studies identified phosphorylation of Dam1c as key event in regulation of Dam1c oligomerization. Phosphorylation of Dam1$^{S20}$ by Ipl1 prevents oligomerization, while Cdk1-dependent phosphorylation of Ask1 (residues S216 and S250) was recently described to promote oligomerization (Wang *et al*, 2007; Zelter *et al*, 2015; Gutierrez *et al*, 2020). However, these studies focused on the intrinsic ability of Dam1c to form large oligomers. With our study, we provide the first evidence that extrinsic factors physically contribute to Dam1c oligomerization. We found that anaphase kinetochore levels of Dad1-GFP were unaffected by deletion of the SxIP motif. This supports the idea that the Dam1c-Bim1 complex represents a metaphase-specific configuration of Dam1c, which is subsequently replaced by other binding partners such as Ndc80c. Based on quantitative live cell fluorescence microscopy, Dhatchinamoorthy *et al* (2017) reported an increase in Ndc80c copy number at anaphase kinetochores compared with metaphase kinetochores, while the number of Dam1c remained unchanged during the cell cycle, suggesting different configurations of Ndc80c, but not of Dam1c during mitosis. Considering our results, we propose that the Dam1c organization at kinetochores only changes regarding its binding partners without affecting the copy number of the complex.

### Crosslinking mass spectrometry and electron microscopy reveals the topology of the Dam1c-Bim1 complex

Mass spectrometry analysis of crosslinked Dam1c in the absence and presence of Bim1 provides important topological information about the Dam1c-Bim1 complex. Our results are in overall agreement with previously published proximity maps of Dam1c (Zelter *et al*, 2015; Legal *et al*, 2016) and a recently published cryo-EM structure of *Chaetomium thermophilum* Dam1c (Jenni & Harrison, 2018). Binding of Bim1 to Dam1c does not change the overall architecture of Dam1c itself since the patterns of crosslinks between Dam1c subunits are similar in the absence and presence of Bim1. Thus, we conclude that binding of Bim1 does not induce a major topological change in the complex. Our crosslinking data support the results of the *in vitro* reconstitution of the Dam1c-Bim1 complex: The Bim1 EBH domain was crosslinked to a lysine residue in proximity to the SxIP motif of Duo1. Electron microscopy of the Dam1c-Bim1 complex places Bim1 in proximity of the protrusion domain, which consists of Spc19 and Spc34 (Jenni & Harrison, 2018). Several crosslinks between Spc19 and Spc34 and Bim1 are in

agreement with this topology (Appendix Table S1). Furthermore, our data confirm close proximity between the C-terminus of Duo1 and both Spc19 and Spc34. The C-terminus of Duo1 itself is not resolved in the cryo-EM structure published by Jenni and Harrison (2018). However, it appears that the C-termini of both Dam1 and Duo1 protrude from adjacent areas of the central region of the T-shaped Dam1c, which allows close proximity of both C-termini with the protrusion domain.

### Phospho-regulation of the Dam1c-Bim1 complex can facilitate the formation of new end-on attachments

Our crosslinking data place the C-terminus of the Dam1 subunit in proximity of the Bim1 binding interface, indicated by multiple cross-links to Duo1 and Bim1. The observation that a C-terminal truncation of Dam1 prevents dissociation of the Dam1c-Bim1 complex under high salt conditions in combination with the crosslinking data leads us to propose that the Dam1 C-terminus acts as intramolecular inhibitor of Bim1 binding (Fig EV2A). The proximity between the Duo1 and Dam1 C-termini may allow partial masking of the Duo1 SxIP motif by the Dam1 C-terminus. Even though the C-termini of Dam1 and Duo1 make the major contribution to microtubule binding of Dam1c (Wang *et al*, 2007; Zelter *et al*, 2015; Legal *et al*, 2016), we find that localization of Dam1c to metaphase kinetochore clusters is increased in a *dam1–19* strain. This is surprising as recombinant Dam1–19c displays lower affinity for microtubules *in vitro* and *dam1–19* strains show a temperature-sensitive phenotype *in vivo* (Cheeseman *et al*, 2001; Westermann *et al*, 2005). We speculate that increased association of Bim1 with Dam1–19c compensates for the impaired microtubule binding of the complex alone. The fact that *dam1–19* is a slow-growing allele shows that it must be compromised in a different aspect. This may be binding to the Ndc80 complex, as suggested in previous studies (Kalantzaki *et al*, 2015), and requires further investigation.

Previous studies implicated Mps1-dependent phosphorylation of Dam1 as an important step for the formation of bi-oriented end-on attachments; however, the underlying molecular mechanism has remained obscure (Shimogawa *et al*, 2006; Maure *et al*, 2007). We show that Mps1 kinase activity can promote binding of Bim1 to Dam1c. We base this conclusion on two observations: First, selective inhibition of Mps1 reduces and overexpression of Mps1 increases binding of Dam1c to Bim1 in pull-down assays (Fig 4E and F, Appendix Fig S3); and second, overexpression of Mps1 alters localization of Dad1-GFP in live cell microscopy and promotes association with microtubules of the mitotic spindle (Fig 4G–I). Whether microtubule localization of Dam1c depends on Bim1 or is mediated by other mechanisms is unclear so far.

We identified Ask1, Duo1, and Bim1 as novel Mps1 substrates (Fig EV3A), and in future studies, mapping the Mps1 phosphorylation sites in these proteins coupled with *in vivo* analysis will help to better understand the role of Dam1c phosphorylation by Mps1. We also note that binding of Dam1c to Bim1 lacking its CH domain and flexible linker (Bim1$^{185–344}$) is increased compared with the full-length protein (Appendix Fig S1A and B) and conclude that the full-length protein adapts a configuration that partially prevents binding of Dam1c. A previous study showed that mimicking Ipl1-dependent phosphorylation of the flexible linker of Bim1 indeed induces a conformation change of Bim1 (Zimniak *et al*, 2009). We propose

that phosphorylation of Bim1 by Mps1 might have a similar effect, which additionally promotes binding to Dam1c.

Based on our findings, we propose that Mps1 kinase regulates multiple aspects of Dam1c function: On the one hand, Mps1 promotes the formation of the Dam1c-Bim1 complex, which facilitates its loading onto kinetochores. On the other hand, phosphorylation of Dam1c by Mps1 appears to negatively regulate Dam1c oligomerization as shown by our biochemistry data, resembling one effect of Ipl1/Aurora B phosphorylation of the complex. Phosphorylation of kinetochore proteins by Ipl1/Aurora B is a key step of error correction and disrupts kinetochore–microtubule attachments lacking tension (Lampson & Cheeseman, 2011). For instance, binding of the Ndc80 complex to Dam1c is weakened by Ipl1/Aurora B-dependent phosphorylation (Lampert *et al*, 2010; Tien *et al*, 2010). Crucially, we show that the Dam1c-Bim1 association is unaffected by Ipl1 phosphorylation of Dam1 (Fig EV4C). This means that even under conditions of high Ipl1 activity, a pool of kinetochore-proximal plus end-localized Dam1c-Bim1 is primed for assembly into a Dam1c ring. Mps1 localizes to unattached kinetochores, phosphorylates kinetochore proteins such as Spc105, Ndc80, and Dam1c (Shimogawa *et al*, 2006; Kemmler *et al*, 2009; London *et al*, 2012), and activates the spindle assembly checkpoint (Musacchio, 2015). Activation of the spindle assembly checkpoint does not only delay anaphase entry until all kinetochores are bi-oriented but also provides mechanisms that actively promote the interaction between kinetochores and microtubules (Sacristan *et al*, 2018). Dam1c and Bim1 localize at the plus ends of dynamic microtubules, while Mps1 exclusively localizes to unattached kinetochores. Only if a growing microtubule plus end with Bim1 and Dam1c comes into proximity to an unattached kinetochore with active Mps1, Dam1c is phosphorylated (Fig 7C). As our data show, this modification promotes binding of Dam1c to Bim1. By this mechanism, Dam1c ring formation is exclusively primed at unattached kinetochores where it is required for formation of stable kinetochore–microtubule attachments, while it is not enhanced at any other site such as microtubules distal from unattached kinetochores. Even though it appears to be contradictory that Mps1 directly prevents Dam1c oligomerization while it indirectly promotes oligomerization by Bim1, our *in vitro* data merely recapitulate a single aspect of error correction and kinetochore–microtubule interactions. The situation *in vivo* is expected to be much more complex integrating Cdk, Ipl1/Aurora B, and Mps1 kinase activities, counteracting phosphatases and microtubule dynamics.

For the first time, we report Dam1c ring formation in the absence of microtubules, highlighting an activity of Bim1 and Bik1 in this process. For several reasons, however, we think that ring formation in cells still depends on microtubules and that it is unlikely that fully assembled rings thread onto a kinetochore. Bim1 and Bik1, the trigger of oligomerization *in vitro*, localize to microtubules *in vivo* (Berlin *et al*, 1990; Schwartz *et al*, 1997; Blake-Hodek *et al*, 2010) with low, if any, free protein in the nucleoplasm. In addition, preassembly of a ring complex followed by random threading of a microtubule appears to be highly inefficient.

### Distinct outer kinetochore configurations allow the stepwise maturation of end-on attachments

Based on our experimental data, we propose that formation of the Dam1c-Bim1 complex has four functions (Fig 7C). First, Dam1c enriches at the microtubule plus end, which enables proximity to the kinetochore. Second, clustering of Dam1c at the microtubule plus end increases its local concentration, which is required for oligomerizing into the characteristic Dam1c ring structure. Bim1 supports this process by simultaneously binding and crosslinking the protrusion domains of adjacent Dam1c decamers, thereby stabilizing oligomers against dissociation. This process is further promoted by the cooperative binding of Dam1c to microtubules (Gestaut *et al*, 2008). Third, Bim1 helps in positioning Dam1c in the correct orientation relative to the kinetochore. Unlike other microtubule-binding proteins, Dam1c does not recognize a specific structural element of microtubules and thus cannot take advantage of the inherent polarity of microtubules (Westermann *et al*, 2006). Jenni and Harrison (2018) propose that the protrusion domain of Dam1c, which is oriented parallel to the microtubule lattice, points toward the microtubule plus end and interacts with the Ndc80 complex. Our structural data of the Dam1c-Bim1 complex reveal that Bim1 binds to the protrusion domain. Since Bim1 preferentially binds the microtubule plus end, Dam1c would be positioned in such a manner that the protrusion domain points toward the kinetochore. Our *in vitro* data further show that binding of Bim1 to Dam1c inhibits Ndc80c binding to Dam1c (Figs 6B and C, and EV5B and C). We propose that Bim1 serves as placeholder, which eventually hands over Dam1c to kinetochore-resident Ndc80c. Fourth, Bim1 recruits Bik1 to Dam1c, which further promotes and stabilizes ring assembly. Notably, in this process Bim1 simultaneously binds two different cargoes: Bik1 via the C-terminal ETF sequence and Dam1c via the EBH domain. A study by Stangier *et al* (2018) using truncated Bik1 and SxIP-containing peptides previously demonstrated that Bim1 can indeed bind two different cargoes at the same time via two distinct mechanism. Here, we show that this is also true in the context of full-length proteins.

Our experimental data reveal a general strategy by which localized kinase activities may be employed to utilize more general MAPs such as Bim1/EB1 for kinetochore functions. It was described previously that the Ska complex, the metazoan functional analog of Dam1c (van Hooff *et al*, 2017), physically interacts with EB1 in human cells (Thomas *et al*, 2016). Perturbation of this interaction results in reduced kinetochore localization of the Ska complex, suggesting that the mechanisms revealed in our study may be more widely applicable. Our study also indicates that +TIPs such as Bim1-Bik1 do not only increase the local concentrations of their cargos at microtubule plus ends, but can also critically contribute to the assembly and organization of large protein complexes at the microtubule plus end. Future studies will have to structurally characterize these distinct kinetochore plus-end configurations in more detail and functionally analyze their contributions to error correction and sister chromatid segregation.

## Materials and Methods

### Purification of recombinant Dam1c from *E. coli*

Recombinant Dam1c was purified from *E. coli* following the protocol described in Westermann *et al* (2005) with the following modifications. Cell lysates were loaded on a HisTrap column (GE Healthcare) installed on an ÄKTA FPLC system. Dam1c was eluted

using a linear imidazole gradient from 20 to 500 mM. Appropriate elution fractions were pooled and subjected to buffer exchange using a HiPrep 26/30 Desalting column (GE Healthcare) before cation exchange chromatography and gel filtration as described in previous publications.

### Purification of recombinant Bim1 and GST-Bim1 from *E. coli*

Recombinant Bim1 and GST-Bim1 and respective truncations were purified from *E. coli* according to the protocol described in Zimniak *et al* (2009) with the following modifications: Cleared cell lysates were loaded on a GSTrap column (GE Healthcare). Elution fractions were collected and subjected to buffer exchange using a HiPrep 26/30 Desalting column. The following purification steps were performed as described in previous publications.

### Purification of recombinant Bik1-6xHis from *E. coli*

Bik1-6xHis was expressed in BL21 Rosetta cells by addition of 0.1 mM IPTG (overnight at 18°C). Cells were resuspended in lysis buffer (20 mM $Na_2HPO_4/NaH_2PO_4$, pH 7.4, 500 mM NaCl, 0.01% (v/v) Triton X-100, 20 mM imidazole, and 1 mM DTT) and lysed by sonication in the presence of 1 mM PMSF (Sigma-Aldrich) and Pierce™ Protease Inhibitor (EDTA-free, Thermo Fisher Scientific). The lysate was cleared by centrifugation and loaded on a HisTrap column (GE Healthcare) installed on an ÄKTA FPLC system. The protein was eluted by a linear imidazole gradient from 20 to 500 mM. Appropriate elution fractions were pooled and subjected to gel filtration (25 mM HEPES, pH 7.4, 350 mM NaCl, 1 mM $MgCl_2$, 2.5% (v/v) glycerol, 0.5 mM TCEP). Glycerol was added to a final concentration of 5% (v/v), and aliquots were snap-frozen in liquid $N_2$.

### Purification of recombinant Mps1 from Sf9 insect cells

Sf9 cells were infected with a recombinant baculovirus carrying an expression construct for Mps1-StrepII. Infected cells were harvested by centrifugation, resuspended in lysis buffer (20 mM $Na_2HPO_4/NaH_2PO_4$, pH 7.4, 300 mM NaCl, 2.5 (v/v) % glycerol, and 0.01 (v/v) % Tween-20 with 1 mM PMSF and Pierce™ Protease Inhibitor (EDTA-free, Thermo Fisher Scientific)), and lysed by sonication. The lysate was incubated at 4°C for 10 min and cleared twice by centrifugation. The lysate was incubated with Strep-Tactin beads (Qiagen) equilibrated in lysis buffer for 4 h of rotating at 4°C. Beads were collected by centrifugation and washed three times with lysis buffer. Mps1-StrepII was eluted by addition of elution buffer (lysis buffer with 2.5 mM desthiobiotin). Glycerol was added to a final concentration of 5% (v/v), and aliquots were snap-frozen in liquid $N_2$.

### Purification of Stu2-Flag from Sf9 insect cells

Sf9 cells were infected with a recombinant baculovirus carrying an expression construct for Stu2-Flag. Infected cells were harvested by centrifugation, resuspended in lysis buffer (20 mM $Na_2HPO_4/NaH_2PO_4$, pH 8.0, 300 mM NaCl, 2.5% (v/v) glycerol, 0.05% (v/v) Tween-20, and 1 mM DTT) with 1 mM PMSF (Sigma-Aldrich) and Pierce™ Protease Inhibitor (EDTA-free, Thermo Fisher Scientific), and lysed by Dounce homogenization. The lysate was cleared by centrifugation at 4°C and incubated with anti-Flag M2 agarose beads (Sigma-Aldrich) for 3 h at 4°C. Beads were collected by centrifugation and washed three times with lysis buffer. Stu2-Flag was eluted from beads by addition of elution buffer (lysis buffer with 0.5 µg/µl 3xFlag peptide). Glycerol was added to a final concentration of 5% (v/v), and aliquots were snap-frozen in liquid $N_2$ and stored at −80°C.

### Purification of Ndc80c-Flag from Sf9 insect cells

Sf9 insect cells were infected with a baculovirus carrying an expression construct for Ndc80c-Flag (Ndc80-Flag, Nuf2, Spc24, and Spc25). Infected cells were harvested by centrifugation, resuspended in lysis buffer (20 mM $Na_2HPO_4/NaH_2PO_4$, pH 6.8, 150 mM NaCl, 5% (v/v) glycerol, 0.05% (v/v) Tween-20, and 0.5 mM TCEP supplemented with Pierce™ Protease Inhibitor [EDTA-free, Thermo Fisher Scientific] and 1 mM PMSF [Sigma-Aldrich]), and lysed by sonication. Lysates were cleared by centrifugation and incubated with anti-Flag M2 agarose beads (Sigma-Aldrich) at 4°C for 3 h. Beads were washed three times with wash buffer (20 mM HEPES, pH 7.4, 200 mM NaCl, 1 mM $MgCl_2$, 5% (v/v) glycerol, and 0.05 mM TCEP), and Ndc80c-Flag was eluted from beads with wash buffer supplemented with 0.5 µg/µl 3xFlag peptide. Protein aliquots were snap-frozen in liquid $N_2$ and stored at −80°C.

### Yeast cultivation and manipulation

All strains were based on yeast strain S288C. Standard techniques were used for cultivation and genetic manipulation. PCR-based integration cassettes were generated according to previously published protocols (Longtine *et al*, 1998; Janke *et al*, 2004; Lee *et al*, 2013). Table 1 lists all plasmids used for yeast transformation. Table 2 contains a list of all strains used in this study.

### Serial dilution assay

Cells from a saturated overnight culture were diluted to $OD_{600} = 0.4$ in minimal medium, and serial 1:4 dilutions were made (six dilutions in total). Dilutions were spotted on YEPD plates, or YEPD plates with 1 µg/ml rapamycin and incubated at different temperatures for 2–4 days.

### Synchronization of yeast cultures using α factor

Yeast cells from an overnight culture were diluted to $OD_{600} = 0.2$ and grown at 30°C for 1 h. α factor was added to a final concentration of 10 µg/ml. Arrest of > 90% of all cells in G1 phase was achieved after about 2 h and was confirmed by microscopy. Cells were harvested by centrifugation, and washed three times in pre-warmed YEPD with 100 µg/ml pronase and once with pre-warmed YEPD without pronase. Cells were finally resuspended in pre-warmed YEPD and cultivated at 37°C. α factor was added about 45 min after release to arrest cells in G1 after one cell cycle.

### Analytical size-exclusion chromatography

Gel filtration was performed with a Superose 6 Increase 3.2/300 column (GE Healthcare) installed on an ETTAN FPLC system. Gel filtration buffer contained 25 mM HEPES, pH 7.4, 200 mM NaCl,

**Table 1.   List of plasmids used for protein expression and construction of yeast strains.**

| Name | Coding sequence | Purpose | Source |
|---|---|---|---|
| pSW1 | Dam1$^{WT}$c with Hsk3-6xHis | Protein purification | Westermann laboratory |
| pSW4 | Dam1-19c with Hsk3-6xHis (Dam1 Q205 Stop) | Protein purification | Westermann laboratory |
| pSW17 | Dam1$^{4D}$c with Hsk3-6xHis (Dam1 S20D, S257D, S265D, S292D) | Protein purification | Westermann laboratory |
| pAD21 | Dam1$^{\Delta SxIP}$c with Hsk3-6xHis | Protein purification | This study |
| pTZ12 | GST-Bim1 | Protein purification | Zimniak et al (2009) |
| pTZ15 | GST-Bim1$^{1-133}$ | Protein purification | Zimniak et al (2009) |
| pTZ13 | GST-Bim1$^{120-344}$ | Protein purification | Zimniak et al (2009) |
| pTZ39 | GST-Bim1$^{185-344}$ | Protein purification | Zimniak et al (2009) |
| pTZ40 | GST-Bim1$^{205-344}$ | Protein purification | Zimniak et al (2009) |
| pET28a-Bik1-6xHis | Bik1-6xHis | Protein purification | Westermann laboratory |
|  | Stu2-Flag | Protein purification | Westermann laboratory |
|  | StrepII-Mps1 | Protein purification | Westermann laboratory |
|  | Ndc80c-Flag | Protein purification | Westermann laboratory |
| pAD4 | Duo1$^{WT}$-6xFlag in pRS306 | Yeast transformation | This study |
| pAD5 | Duo1$^{\Delta SxIP}$-6xFlag in pRS306 | Yeast transformation | This study |
| pDD526 | Pds1–18xMyc | Yeast transformation | Westermann laboratory |
| pAD19 | dam1–19 in pRS305 | Yeast transformation | This study |
| pAD24 | Dam1$^{WT}$ in pRS305 | Yeast transformation | This study |
| pSW809 | Ndc80$^{WT}$ in pRS306 | Yeast transformation | Lampert et al (2013) |
| pAD67 | Ndc80$^{\Delta 490-510}$ in pRS306 | Yeast transformation | This study |
| pAD56 | Ask1$^{WT}$ in pRS303 | Yeast transformation | This study |
| pAD72 | Ask1$^{2A}$ (S216A, S250A) in pRS303 | Yeast transformation | This study |
| pAD73 | Ask1$^{2D}$ (S216D, S250D) in pRS303 | Yeast transformation | This study |

1 mM MgCl$_2$, 0.5 mM TCEP, and 2.5 (v/v) % glycerol. Experiments combining Dam1c, Bim1, Bik1, and Ndc80c were performed in the same buffer with 150 mM NaCl. For indicated experiments, the NaCl concentration was increased to 400 mM. Elution fractions of 100 µl were collected and analyzed by either SDS–PAGE and stained with Coomassie Brilliant Blue (Fairbanks et al, 1971) or by Western blot.

Gel Filtration Standard was purchased from Bio-Rad (catalog number 1511901).

**Preparation of soluble yeast cell extracts**

Soluble cell lysates were used in pull-down assays. To test for the effect of temperature-sensitive alleles, strains were cultivated at 30°C for 2 h and then shifted to 37°C for 4–5 h. For expression of pGal-1xMyc-Mps1 from the Gal promoter, strains were grown in YEP with 2% raffinose for 2 h at 30°C. Galactose was added to a final concentration of 2%, and the strains were cultivated at 30°C for 4 h. For selective inhibition of Mps1, cells carrying the mps1-as1 allele (Jones et al, 2005) were treated with 10 µM 1NM-PP1 for 3.5 h. Cells were harvested by centrifugation, washed in 1 ml H$_2$O, and resuspended in 700 µl lysis buffer (25 mM HEPES, pH 8.0, 150 mM NaCl, 2 mM EDTA, 5 (v/v) % glycerol, and 1 mM DTT). Pierce$^{TM}$ Protease Mini Inhibitor and phosphatase inhibitor mixes (Thermo Fisher Scientific) and 1 mM PMSF (Sigma-Aldrich) were added according to the manufacturer's instructions. Cells were disrupted by bead beating (three times 30 s, BioSpec Mini-Beadbeater). NP-40 was added to a final concentration of 0.1 (v/v) %, and the lysates were incubated at 4°C for 10 min of rotating. The beads were briefly spun down, and the supernatant was transferred to a new tube. The lysate was cleared by centrifugation in a tabletop centrifuge at full speed for 20 min at 4°C.

**Crosslinking mass spectrometry**

For titration of the crosslinker, an equimolar mixture of isotopically light- and heavy-labeled BS3-H12/D12 (bis(sulfosuccinimidyl)suberate; Creative Molecules) was diluted to different concentrations ranging from 0 to 2,500 µM. In a total reaction volume of 6 µl, 1 µg of Dam1c was mixed with BS3 of different concentrations and incubated at 35°C for 45 min. The reaction was stopped by addition of NH$_4$HCO$_3$ to a final concentration of 100 mM. The samples were subsequently analyzed by SDS–PAGE followed by silver staining. For mass spectrometric analysis of crosslinked Dam1c, 60 µg of Dam1c was crosslinked in the absence or presence of 7.4 µg Bim1 using a BS3 concentration of 600 µM. The crosslinked samples were processed, and the crosslinked peptides were identified as previously described (Herzog et al, 2012). The crosslink-derived distance restraints were visualized using the free software xVis (Grimm et al, 2015).

**FACS analysis of DNA content of yeast cells**

0.5 ml of a yeast culture was harvested by centrifugation and resuspended in 300 µl H$_2$O. 700 µl ethanol was added dropwise and incubated overnight. Cells were spun down, resuspended in 1 ml Tris buffer (50 mM Tris, pH 7.5), sonicated in a water bath, spun down again, resuspended in 1 ml Tris buffer with 0.2 mg/ml RNase (Roche), and incubated at 50°C for 2 h. 50 µl of 20 mg/ml Proteinase K (Roth) was added and further incubated at 50°C for 2 h. The cells were collected by centrifugation and resuspended in 500 µl Tris buffer. 50 µl of the cell suspension was mixed with 450 µl of

**Table 2.  List of yeast strains used in this study.**

| Strain name | Genotype | Source | Figure |
|---|---|---|---|
| DDY902 | Mat a, his3Δ200, ura3-52, ade2-1, leu2-3,112 | Westermann laboratory | Appendix Fig S1, Fig 3 |
| DDY904 | Mat α, his3Δ200, ura3-52, leu2-3,112, lys2-801 | Westermann laboratory | Appendix Fig S2 |
| ADY5 | Mat a, Duo1$^{WT}$-6xFlag::Leu2, his3Δ200, ura3-52, ade2-1 | This study | Figs 2 and 4 |
| ADY8 | Mat a, Duo1$^{ΔSxIP}$-6xFlag::Leu2, his3Δ200, ura3-52, lys2-801 | This study | Fig 2 |
| ADY14 | Mat a, duo1Δ::His3, leu2-3,112::Duo1$^{ΔSxIP}$-6xFlag::Leu2, ura3-52, ade2-1 | This study | Appendix Fig S1 |
| ADY17 | Mat a, Duo1$^{WT}$-6xFlag::KanMx6, his3Δ200, ura3-52, ade2-1, leu2-3,112 | This study | Appendix Fig S1 |
| ADY20-2 | Mat a, Duo1$^{ΔC(R223Stop)}$-6xFlag::KanMx6, his3Δ200, ura3-52, lys2-801, ade2-1, leu2-3,112 | This study | Appendix Fig S1 |
| ADY29 | Mat a, Pds1-18xMyc::Leu2, his3Δ200, ura3-52, ade2-1 | This study | Fig 3 |
| ADY226 | Mat a, duo1Δ::His3; leu2-3,112::Duo1$^{WT}$-6xFlag::Leu2; Pds1-18xMyc::Leu2, ura3-52, lys2-801, ade2-1 | This study | Fig 3 |
| ADY227 | Mat a, duo1Δ::His3; leu2-3,112::Duo1$^{ΔSxIP}$-6xFlag::Leu2; Pds1-18xMyc::Leu2; ura3-52, ade2-1 | This study | Fig 3 |
| ADY240 | Mat a, Pds1-18xMyc::Leu2, his3Δ200, ura3-52, lys2-801 | This study | Fig 3 |
| ADY231 | Mat a, Dad1-GFP::KanMx6; Pds1-18xMyc::Leu2, his3Δ200, ura3-52, lys2-801 | This study | Fig 3 |
| ADY232 | Mat a, Dad1-GFP::KanMx6; Pds1-18xMyc::Leu2; duo1Δ::His3; leu2-3,112::Duo1$^{WT}$-6xFlag::Leu2, ura3-52, lys2-801 | This study | Fig 3, Appendix Fig S2 |
| ADY233 | Mat a, Dad1-GFP::KanMx6; Pds1-18xMyc::Leu2; duo1Δ::His3; leu2-3,112::Duo1$^{ΔSxIP}$-6xFlag::Leu2, ura3-52, lys2-801 | This study | Fig 3, Appendix Fig S2 |
| ADY66 | Mat a, duo1Δ::His3; leu2-3,112::Duo1$^{ΔSxIP}$-6xFlag::Leu2; Dad1-GFP::KanMx6, ura3-52 | This study | Fig 3 |
| ADY72 | Mat a, Duo1-6xFlag::KanMx6; Dad1-GFP::His3, ura3-52, lys2-801, leu2-3,112 | This study | Fig 3 |
| ADY69 | Mat α, Dad1-GFP::His3, ura3-52, lys2-801, leu2-3,112 | This study | Fig 4, Appendix Fig S2 |
| ADY98 | Mat α, *dam1-19*::Ura3; Dad1-GFP::His3, leu2-3,112 | This study | Fig 4 |
| ADY288.1 | Mat a, Duo1$^{WT}$-6xFlag::Leu2; ura3-52::pGal-1xMys-Mps1::Ura3, his3Δ200, ade2-1, trp1-1 | This study | Fig 4 |
| ADY68 | Mat a, Dad1-GFP::His3, ura3-52, leu2-3,112 | This study | Appendix Fig S2 |
| ADY349 | Mat α, Dad1-GFP::His3, Ndc80-mRuby2::KanMx6, ura3-52, lys2-801, leu2-3,112 | This study | Fig 4 |
| ADY351 | Mat α, Dad1-GFP::His3, Ndc80-mRuby2::KanMx6, ura3-52::pGal-1xMyc-Mps1::Ura3, lys2-801, leu2-3,112 | This study | Fig 4 |
| ADY356 | Mat a, Dad1-GFP::His3, Nup60-RedStar2::natNT2, ura3-52, leu2-3,112 | This study | Fig EV2 |
| ADY357 | Mat a, Dad1-GFP::His3, Nup60-RedStar2::natNT2, ura3-52::pGal-1xMyc-Mps1::Ura3, lys2-801, leu2-3,112 | This study | Fig EV2 |
| ADY290 | Mat a, Dad1-GFP::His3, *bim1Δ*::KanMx6, ura3-52, leu2-3,112 | This study | Appendix Fig S2 |
| ADY312 | Mat α, ura3-52::pGal-osTir::Ura3, Dad1-GFP::His3, Cdc20-AID-9xMyc::KanMx6, lys2-801, leu2-3,112 | This study | Fig EV2 |
| ADY263 | Mat α, tor1-1 fpr1::natNT2, RPL13A-2xFKBP12::Trp1, Ndc80-FRB-GFP::KanMX6, Duo1$^{WT}$::His3 | This study | Fig 6 |
| ADY264 | Mat α, tor1-1 fpr1::natNT2, RPL13A-2xFKBP12::Trp1, Ndc80-FRB-GFP::KanMX6, Duo1$^{ΔSxIP}$::His3 | This study | Fig 6 |
| ADY394 | Mat α, tor1-1 fpr1::natNT2, RPL13A-2xFKBP12::Trp1, Ndc80-FRB-GFP::KanMX6, Duo1$^{WT}$::His3, leu2-3,112::Ndc80$^{WT}$::Leu2 | This study | Fig 6 |
| ADY395 | Mat α, tor1-1 fpr1::natNT2, RPL13A-2xFKBP12::Trp1, Ndc80-FRB-GFP::KanMX6, Duo1$^{WT}$::His3, leu2-3,112::Ndc80$^{Δ490–510}$::Leu2 | This study | Fig 6 |
| ADY396 | Mat α, tor1-1 fpr1::natNT2, RPL13A-2xFKBP12::Trp1, Ndc80-FRB-GFP::KanMX6, Duo1$^{ΔSxIP}$::His3, leu2-3,112::Ndc80$^{WT}$::Leu2 | This study | Fig 6 |
| ADY397 | Mat α, tor1-1 fpr1::natNT2, RPL13A-2xFKBP12::Trp1, Ndc80-FRB-GFP::KanMX6, Duo1$^{ΔSxIP}$::His3, leu2-3,112::Ndc80$^{Δ490–510}$::Leu2 | This study | Fig 6 |
| ADY436 | Mat a, Duo1$^{WT}$-6xFlag::Leu2, Ask1$^{WT}$::His3, ura3-52, ade2-1 | This study | Fig 7 |
| ADY438 | Mat a, Duo1$^{WT}$-6xFlag::Leu2, Ask1$^{2A}$::His3, ura3-52, ade2-1 | This study | Fig 7 |
| ADY440 | Mat a, Duo1$^{WT}$-6xFlag::Leu2, Ask1$^{2D}$::His3, ura3-52, ade2-1 | This study | Fig 7 |
| ADY450 | Mat a, Duo1$^{ΔSxIP}$-6xFlag::Leu2, Ask1$^{WT}$::His3, ura3-52, ade2-1 | This study | Fig 7 |
| ADY452 | Mat a, Duo1$^{ΔSxIP}$-6xFlag::Leu2, Ask1$^{2A}$::His3, ura3-52, ade2-1 | This study | Fig 7 |

**Table 2** (continued)

| Strain name | Genotype | Source | Figure |
|---|---|---|---|
| ADY453 | Mat α, Duo1$^{\Delta SxIP}$-6xFlag::Leu2, Ask1$^{2A}$::His3, ura3-52, lys2-801 | This study | Fig 7 |
| ADY454 | Mat a, Duo1$^{\Delta SxIP}$-6xFlag::Leu2, Ask1$^{2D}$::His3, ura3-52, ade2-1 | This study | Fig 7 |
| DDY1502 | Mat a, *mad1Δ*::His3, his3Δ200, leu2-3,112, ura3-52, ade2-101 | Westermann laboratory | Appendix Fig S2 |
| SWY754 | Mat a, *bim1Δ*::KanMx6, his3Δ200, ura3-52, ade2-1, leu2-3,112 | Westermann laboratory | Appendix Fig S2 |
| ADY197 | Mat α, mps1Δ::KanMx6, Trp1::10xMyc-mps1-as1, ipl1-2, Duo1$^{WT}$-6xFlag::KanMx6, ura3-52 | This study | Appendix Fig S3 |
| ADY142 | Mat a, Duo1$^{WT}$-6xFlag::KanMx6, mps1-737, his3Δ200, ura3-52, lys2-801, leu2,3-112 | This study | Fig 4 |

SYTOX Green solution (1 μM SYTOX Green [Invitrogen] in Tris buffer) and incubated for 30 min at room temperature. Cells were analyzed with MACSQuant VYB flow cytometer (Miltenyi Biotec), and data were processed using FlowJo.

### Yeast live cell microscopy

Cells from an overnight culture were diluted to OD$_{600}$ = 0.2 in synthetic defined medium with 2% glucose lacking tryptophan (SD doTrp + glucose) and grown at 30°C for 4 h. Cells were briefly sonicated and immobilized on a glass bottom dish (MatTek Corporation) using concanavalin A. For microscopy of strains with pGal-1xMyc-Mps1, cells were initially grown in SD doTrp medium with 2% raffinose. Galactose was added after 2 h to a final concentration of 2%, and cells were grown for another 4 h.

All microscopy was performed at 30°C with a DeltaVision Elite wide-field microscope (GE Healthcare Life Sciences) with a Plan Apo 60x/1.42 oil objective, solid state light source (SSCI), and sCMOS camera. Deconvolution of images was acquired using standard settings.

The free software ImageJ was used for quantification of Dad1-GFP signal intensities at kinetochore clusters. The raw integrated density in equally sized areas was measured and corrected for cellular background signal.

### Statistical analysis

The software GraphPad Prism was used for statistical analysis of acquired data. Unless otherwise stated, the mean ± standard deviation (SD) are displayed for the quantification of signal intensities at kinetochore clusters. All values were normalized to the mean signal intensity at metaphase cluster of the corresponding wild-type strain. Sample size and applied statistical test are described in the corresponding figure legends.

### Pull down assays

For pull-down assays using recombinant Dam1c, BSA-blocked glutathione sepharose beads (GE Healthcare) were loaded with 6 μg GST or 15 μg GST-Bim1$^{185-344}$. Dam1c was diluted to 5 μM in interaction buffer (25 mM HEPES, pH 7.4, 300 mM NaCl, 1 mM MgCl$_2$, 5 (v/v) % glycerol, 0.5 mM TCEP, 2 mM EDTA, and 0.1 (v/v) % Triton X-100), 120 μl was added to the loaded beads, and the reaction was incubated for 15 min at 4°C of rotating. Beads were

subsequently washed three times for 5 min in interaction buffer and finally boiled in 50 μl SDS loading dye, and the supernatant was loaded for SDS–PAGE.

For pull-down assays with soluble yeast cell lysates, BSA-blocked glutathione sepharose beads (GE Healthcare) were loaded with 9 μg GST or 45 μg GST-Bim1 or GST-Bim1$^{185-344}$. 300–400 μl of cell lysate with a protein concentration of 2 μg/μl was added to the beads and incubated at 4°C for 15 to 120 min. Beads were collected by centrifugation and washed three times for 5 min in wash buffer (25 mM HEPES, pH 7.4, 300 mM NaCl, 1 mM MgCl$_2$, 5 (v/v) % glycerol, 0.5 mM TCEP, 2 mM EDTA, and 0.1 (v/v) % Triton X-100). Beads were boiled in 50 μl SDS loading dye, and the supernatant was loaded for SDS–PAGE.

For pull-down assays with Ndc80c-Flag, anti-Flag M2 agarose beads were incubated with cell lysate from Sf9 insect cells expressing Ndc80c-Flag (with Flag-tagged Ndc80). After loading of beads with Ndc80c-Flag, beads were washed with interaction buffer (25 mM HEPES, pH 7.4, 200 mM NaCl, 1 mM MgCl$_2$, 2.5% (v/v) glycerol, 0.5 mM TCEP, and 0.01% (v/v) Triton X-100) and Dam1c or Bim1 alone or Dam1c pre-incubated with increasing amounts of Bim1 was added to the beads and incubated at 4°C for 15 min. Beads were washed three times with interaction buffer, and proteins were eluted from beads with interaction buffer supplemented with 0.5 μg/μl 3xFlag peptide. Samples were subsequently analyzed by SDS–PAGE.

For Stu2-Flag pull-down assays, anti-Flag M2 agarose beads were incubated with lysates of Sf9 cells expressing Stu2-Flag. Beads were washed three times with interaction buffer (25 mM HEPES, pH 7.4, 200 mM NaCl, 1 mM MgCl$_2$, 2.5% (v/v) glycerol, 0.5 mM TCEP, and 0.01% (v/v) Triton X-100) and incubated with buffer, Dam1c, or Bik1 at 4°C for 15 min. Beads were subsequently washed three times with interaction buffer, and bound proteins were eluted from the beads with elution buffer (interaction buffer supplemented with 0.5 μg/μl 3xFlag peptide). To test for unspecific binding of Dam1c or Bik1 to the beads, BSA-blocked beads were incubated with either Dam1c or Bim1. Input and elution samples were analyzed by SDS–PAGE.

### *In vitro* kinase assay

*In vitro* kinase reactions were prepared in a total reaction volume of 20 μl. 6.75 μg Dam1c or 1.5 μg Bim1 was diluted in kinase buffer (20 mM HEPES, pH 7.5, 100 mM KCl, 10 mM MgCl$_2$, 10 mM MnCl$_2$, 25 mM β-glycerol phosphate, and 1 mM DTT) supplemented with 50 μM ATP and 1 μl $^{32}$P-γ-ATP (10 μCi/μl; Hartmann Analytic) in the presence or absence of recombinant Mps1. Samples were incubated at

30°C for 30 min. The reaction was stopped by addition of 6x SDS loading dye, and samples were loaded on an SDS–polyacrylamide gel. The incorporation of $^{32}P$ was detected by autoradiography.

For *in vitro* phosphorylation prior to analytical size-exclusion chromatography, Dam1c, Bim1, or Dam1c and Bim1 was diluted in kinase buffer (20 mM HEPES, pH 7.5, 100 mM KCl, 10 mM MgCl$_2$, 25 mM β-glycerol phosphate, and 1 mM DTT), mixed with recombinant Mps1, and incubated in the absence or presence of 500 μM ATP at 30°C for 2 h.

### Antibodies

The following antibodies were used for detection of proteins in Western blots: mouse monoclonal HRP-conjugated anti-Flag M2 antibody (Sigma), mouse monoclonal anti-GFP antibody (Roche), goat polyclonal anti-GST antibody (GE Healthcare), mouse monoclonal HRP-conjugated anti-penta-His antibody (Qiagen), mouse monoclonal anti-Myc 9E10 antibody (Covance), and mouse monoclonal anti-Pgk1 antibody (Invitrogen).

### Quantification of Western blot signals

Raw integrated densities of Pds1 and Pgk1 were measured using Fiji and were corrected for local background signal. Pds1 signals were divided by the values of the corresponding Pgk1 signal. Values were finally normalized by setting the highest value as 1.00.

### Multiple sequence alignment

Sequences of homologues protein sequences were collected from the Fungal Orthogroups Repository (Broad Institute). Alignments were generated using MAFFT and processed and visualized using Jalview (Clustalx coloring scheme with 50% conservation threshold).

### Electron microscopy of Dam1c, Dam1c-Bim1, and Dam1c-Bim1-Bik1

0.8 μM Dam1c without or with 1.5 μM Bim1 was subjected to size-exclusion chromatography and crosslinked using 0.5% glutaraldehyde. The crosslinking reaction was stopped after 60 s by the addition of Tris (Figs 1C and D, and EV1D and E). Samples were further processed as described in the following section.

4 μl of Dam1c, Dam1c-Bim1, or Dam1c-Bim1-Bik1 complex at concentrations of 0.005–0.05 mg/ml (the initial molar ratios of Dam1c:Bim1:Bik1 were 3:1:2.5 prior to gel filtration) was applied on freshly glow-discharged carbon-coated copper grids (Agar Scientific, G400C). For EM analysis at low salt conditions (Fig 5D and E), the complexes were diluted to buffer with a final salt concentration of 10 mM NaCl before the analysis. After an incubation of 2 min, the sample was blotted with Whatman no. 4 filter paper, washed two times with ddH$_2$O, and stained with 0.75% uranyl formate. Negatively stained proteins were imaged with a JEOL JEM-1400 TEM equipped with a LaB$_6$ cathode operating at 120 kV. Digital electron micrographs were recorded with a 4k × 4k CMOS Camera F416 (TVIPS) under minimal dose conditions with a pixel size of 1.36 Å. A total of 28,140 single particles for Dam1c and 36,980 single particles for crosslinked Dam1c-Bim1 were boxed using crYOLO (Wagner *et al*, 2019) after training it with 10 manually boxed micrographs.

Reference-free classification of the particles was carried out with the iterative stable alignment and clustering approach (Yang *et al*, 2012) (ISAC) implemented in the SPIRE software (Moriya *et al*, 2017). An initial 3D structure was calculated from the best 2D class averages (2D class averages of monomers were excluded at this stage) with the validated ab initio 3D structure determination approach VIPER (Moriya *et al*, 2017) implemented in SPHIRE. CHIMERA (Pettersen *et al*, 2004) was used for visualization.

## Data availability

The mass spectrometry proteomics data have been deposited to the ProteomeXchange Consortium via the PRIDE (Perez-Riverol *et al*, 2019) partner repository (https://www.ebi.ac.uk/pride/archive) with the dataset identifier PXD026935 (http://www.ebi.ac.uk/pride/archive/projects/PXD026935).

The electron microscopy data from this study have been deposited to the Electron Microscopy Data Bank (EMDB, https://www.ebi.ac.uk/emdb) and assigned the identifier EMD-13152 (http://www.ebi.ac.uk/emdb/entry/EMD-13152; Dam1c) and EMD-13151 (http://www.ebi.ac.uk/emdb/entry/EMD-13151; Dam1c-Bim1), respectively.

Expanded View for this article is available online.

## Acknowledgements

A.D. and S.W. thank Kerstin Killinger for support with graphical illustrations. A.D. and S.W. received funding from the German Research Foundation (DFG) in the context of the Collaborative Research Center CRC1093 "Supramolecular chemistry on proteins". C.G. is grateful to Stefan Raunser for the continuous support. S.W. and C.G. acknowledge support by the DFG Collaborative Research Center CRC1430 "Molecular Mechanisms of Cell State Transitions". F.H. was supported by the European Research Council (ERC-StG no. 638218), the Human Frontier Science Program (RGP0008/2015), the Research Training Center (GRK1721) of the German Research Foundation, and the Bavarian Research Center of Molecular Biosystems. Live cell microscopy was supported by the Imaging Center Campus Essen (ICCE) core facility. The DeltaVision Elite high-resolution microscope was obtained through DFG funding (Major Research Instrumentation Programme as per Art. 91b GG, INST 20876/275-1). Open Access funding enabled and organized by Projekt DEAL.

## Author contributions

AD conducted and analyzed the majority of experiments. LE, CB, and BUK prepared and optimized samples for electron microscopy studies. LE, CB, BUK, and CG analyzed and processed electron microscopy data. NK prepared reagents and performed initial SEC analysis of Dam1c assemblies. KJ performed FACS analysis of yeast cells. PR and FH processed and analyzed the crosslinking samples. AD and SW designed the study and wrote the manuscript.

### Conflict of interest

The authors declare that they have no conflict of interest.

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
