## [Review Process File · The EMBO Journal]

Phospho-regulated Bim1/EB1 interactions trigger Dam1c ring assembly at the budding yeast outer kinetochore

Stefan Westermann, Alexander Dudziak, Lena Engelhard, Cole Bourque, Björn Klink, Pascaline Rombaut, Nikolay Kornakov, Karolin Jänen, Franz Herzog, and Christos Gatsogiannis
DOI: [10.15252/embj.20211108004](https://doi.org/10.15252/embj.20211108004)

Corresponding authors: Stefan Westermann (stefan.westermann@uni-due.de)

Review Timeline:

Submission Date:	12th Feb 21
Editorial Decision:	17th Mar 21
Revision Received:	31st May 21
Editorial Decision:	23rd Jun 21
Revision Received:	1st Jul 21
Accepted:	7th Jul 21

Editor: Hartmut Vodermaier

Transaction Report:

Thank you for submitting your manuscript on Dam1c ring assembly for our consideration. It has now been reviewed by three expert referees, whose comments are copied below. As you will see, all referees consider your study interesting and well-conducted, and would therefore be supportive of publication after satisfactory addressing a number of specific, often presentational points. This includes the issue of certain overstatements as well as the request for further discussion of particular aspects; while additional EM experiments as encouraged as an optional point by referee 3 would in my view not be necessary within the scope of this revision.

Referee #1:

Dudziak et al use a combination of biochemistry, cross-linking mass spectrometry, single particle electron microscopy (EM), cell imaging, and genetic techniques to examine the interaction between the budding yeast +TIP, Bim1, and the Dam1c kinetochore complex. Dam1c is well-known as a component of the yeast outer kinetochore crucial for stabilizing attachments between chromosomes and microtubules in the mitotic spindle. In vitro, Dam1c oligomerizes into striking microtubule-encircling rings that enable kinetochores to make strong yet dynamic attachments to growing and shrinking microtubule ends. Bim1, like its homologue EB1, famously binds and tracks growing microtubule plus ends. A prior two-hybrid screen suggested that Bim1 and Dam1c might bind to one another but the relevance of their interaction was unexplored. This new study presents the first analysis of the subdomains involved in the Bim1-Dam1c interaction and provides an initial

view of the structural arrangement. The data are high quality, novel, and provide support for an interesting potential role for the Bim1-Dam1c interaction as a means to orient the Dam1c for proper loading onto its kinetochore receptor, Ndc80c. Unfortunately, however, the implications are overstated in at least two ways, listed below. In my opinion the authors need to deal with these overinterpretations before the manuscript is ready for publication.

It is asserted at least three times in the manuscript that the binding of Bim1 or Ndc80c to Dam1 is mutually exclusive. (E.g., in the section heading on line 422. Also on lines 440-441. Also on lines 656-657.) No data presented here can support this claim. A mere 30% reduction in the amount of Dam1c pulled down by immobilized Ndc80-FLAG was observed after addition of 16 μ M Bim1 (Figure 6B and C). This observation indicates that Bim1 weakly inhibits the Dam1c-Ndc80c interaction. A bulk pull-down experiment like this is insufficient to test for mutual exclusivity.

It is also asserted that Mps1-dependent phosphorylation of Dam1c "increases the affinity of Dam1c for Bim1" (lines 519-520; a similar assertion is also made on lines 355-356). This conclusion is only indirectly supported, by decreased/increased pull-down of Dam1c from clarified lysates from cells with mutant/over-expressed Mps1, using an immobilized fragment of Bim1 as bait. Since Mps1 phosphorylates many targets in cells, these differences are not necessarily due to changes in the biochemical affinity of Dam1c for Bim1.

Referee #2:

The MS of Dudziak et al explores how plus-end associated proteins regulate the assembly of Dam1c as rings at attached kinetochores. A combination of biochemistry, genetics, and structural analysis are nicely combined to tease out the interactions between Bim1 (EB1 ortholog), Bik1, Dam1c, Ndc80c, and Mps1. It shows that Bim1 alone promotes oligomerization of Dam1c as partial rings and that Bim1 plus Bik1 promotes Dam1c's oligomerization into complete rings. This interaction is enhanced by Mps1, which, due to its localization at unattached kinetochores, suggests a model in which Mps1 facilitates Dam1 ring formation at the kinetochore microtubule plus end that has just attached. Through the identification of a possible Bim1:Dam1c interface, the MS also proposes that Bim1, which is localized to microtubule plus ends, can select for Dam1c that is oriented with its protrusion pointed toward the microtubule plus end. This model also explains how Dam1c, which by itself does not appear to sense microtubule polarity, is able to assemble with the kinetochore in the correct orientation. I think this MS contributes important advances to the kinetochore field and suggests exciting follow-up studies, particularly a high-resolution characterization of the newly reconstituted Bim1-Dam1c and Bim1-Bik1-Dam1c complexes.

I only have minor concerns, which if addressed should make the MS clearer to a broader readership.

Line 43 "Faithful and error-free chromosome segregation" - The wording seems redundant. Isn't error-free segregation already faithful?

Line 126 "This result shows that Bim1 is a specific binding partner of Dam1c at microtubule plus ends" - While the experiments in this paragraph do show that Bim1 and Dam1c bind specifically, they don't show that they bind at the MT plus ends. To support such a claim, one would have to show that the complex forms on MT plus ends, either in vitro or in situ. If this statement is based on light microscopy co-localization data, then the relevant papers should be

cited and the strong term "shows" replaced with the more tenuous term "suggests".

Line 157 "Bim1 was visible as an additional mass of approximately 150 kDa" - How was the mass of the putative Bim1 density estimated from the class average? What is the basis for the conclusion that the extra density at the protrusion domain is a Bim1 homodimer? There seems to be a circular argument going on here. Is there additional evidence from the data, which may not be obvious to non-structural-biology readers?

Lines 170-174 "In contrast, the Dam1c-Bim1 complex showed larger oligomers than Dam1c alone, with a curvature that is consistent with early steps of ring formation." - This statement implies that the curvature of Dam1c multimers during early ring formation is known. Please clarify.

Lines 145-174 - Please comment on the confusing relationship between the apparent masses of the various complexes, seen by gel filtration versus by EM. Gel filtration shows that Dam1c alone elutes as a mono-modal species larger than 670 kDa whereas the EM suggests a mixture of monomers and dimers. When Bim1 is added, Gel filtration still shows a monomodal species whereas the EM shows a shift to the Dam1c dimer. The Dam1c monomer is ~ 200 kDa and a Bim1 monomer is ~ 75 kDa, so a 2:2 complex of Dam1c:Bim1 would be 550 kDa. Should we conclude from the chromatogram in Fig. 1A that Dam1c alone in solution is actually a multimer and that when subjected to EM sample preparation, dissociates into monomers unless stabilized by Bim1?

Lines 234 "Strikingly, the kinetochore clusters tumbled in the nucleus" - The nucleus is not labelled in this experiment, so one cannot exclude the possibility that it's the nucleus that's tumbling around inside the cell. A tumbling nucleus would hint at changes in the cytoplasmic portion of the cytoskeleton, in addition to the spindle.

Line 250: "substantial Pds1 were still detectable". Perhaps quantifying the western blots at Fig. 3E would help show the "substantial" difference in Pds1 levels for WT versus Duo1[delta]SxIP strain?

Lines 347-349 and Fig. 4G (see below) I don't see the "elongated structures" indicated by the white arrows.

Lines 406-407 Replace "degradation" with "dissociation".

Lines 408-409 Replace "and have a relatively thick circumference" with "and are thicker than rings assembled in the presence of Bim1 alone"

Line 541 The notion of a metaphase-specific configuration of Dam1c is an interesting one and should be briefly discussed relative to an earlier study (Dhatchinamoorthy et al "Structural plasticity of the living kinetochore."), which showed evidence that Ndc80c has a anaphase configuration while Dam1c did not, as assessed by copy numbers in live-cell fluorescence imaging.

Lines 635-636 "Thus, ring formation is prevented at any other site such as already attached kinetochores or microtubules distal from unattached kinetochores." - This sentence is poorly worded. It makes it sound like Dam1c rings are prevented from forming at the already attached kinetochores. It's probably safer to say that "ring-formation is not enhanced at any other site".

Lines 646-656 "Since Bim1 preferentially binds the microtubule plus end, Dam1c would be positioned in such a manner that the protrusion domain points towards the kinetochore" - This statement makes assumptions about the orientation Bim1's Dam1c-binding domain when it's on

the microtubule. It assumes that Bim1's Dam1c binding domain is oriented parallel to the MT surface, pointing toward the plus end. If, instead, Bim1's Dam1c domain is oriented perpendicular to the MT's surface, then the proposed model wouldn't make sense, and would require additional modifications such as a conformational change upon Dam1c binding. Is there any evidence that this interface is indeed oriented toward the minus end?

OTHER POINTS

Please clarify what "+TIP" acronym means, for non-expert readers.

The term "Negative stain EM images of X" is inaccurate. It should be "EM images of negatively stained X".

The phrase "Interestingly/strikingly ..." is used too often. It implies surprise at a newly presented result when in most cases, the result was fully consistent with those that came before the term interestingly/strikingly. If "Consistent / In agreement with the previous result ..." is substituted for "strikingly", I believe the MS will make more sense. If the results are indeed striking/interesting, please explain why.

The MS reports that overexpression of Mps1 affects the localization of Dad1-GFP in metaphase. This is characterised by the increase in percentage of cells showing bar-shaped Dad1-GFP localisation (Fig. 4G, H, I). Could the formation of this bar-shaped morphology simply be the result of excess accumulation Dad1-GFP / Dam1c? Fig. 4H seems to indicate that the overall Dad1-GFP intensity is higher when Mps1 was overexpressed.

Lines 638-85 + Fig. 5D,E + Fig. 7C: The in vitro experiments show that it is possible to assemble complete rings without microtubules. Please speculate on what this finding means in vivo: Is it possible that inside cells, complete rings form first and then thread on the kinetochore microtubules? Or are their conditions that ensure ring formation only happens in the context of microtubules? I believe this MS provides enough evidence against the former possibility, due to the localization of Bim1 on microtubules in vivo. Though one cannot rule that that there are sufficient copies of soluble Bim1, Bik1, and Dam1c to form rings in nucleoplasm, independently of the spindle.

FIGURES

In the chromatogram of Fig. S1A, the positions of the Dam1c (alone) and Stu2 (alone) peaks are awfully close. Is there a co-IP experiment to better support the conclusion that Stud2 and Dam1c don't interact in solution?

The legends for Fig. 1A and S1A don't say what the gels correspond to. Based on the color scheme, they are presumably fractions collected from the chromatography runs. It would be good to say so in the legend.

Fig. 1D: I suggest replacing the blue and yellow ovals with dashed oval lines. The current rendering makes it look like the relevant features are behind the density map.

Fig. 3E: Why was the Dad1-GFP background strain used for the cell cycle progression experiments? In line 221, it's stated that the Dad1-GFP + Duo1[delta]SxIP suffer severe growth defects at 37C,

compared to strain that expresses untagged Dad1n. I think doing these experiments (Pds1 western blots and FACS) on an untagged Dad1 background strain would better support the conclusion at line 525 that Duo1[delta]SxIP allele causes delayed cell cycle progression.

Fig. 4G, lower panels: the arrowheads don't seem to be pointing to anything.

Fig. S4E - In the pGal-Mps1 cell, the nucleus appears bigger and the Nup signals appear more dispersed. Could this difference be a side-effect of Mps1 overexpression?

Lu Gan

Referee #3:

In this study, the authors found that yeast +TIP protein Bim1 directly binds the Dam1 complex (Dam1c). This binding is mediated by the SxIP motif at the Duo1 C-terminus. With the duo1 mutant lacking SxIP, the amount of the Dam1c is reduced at kinetochores in yeast cells and metaphase is prolonged. Moreover, the Dam1c-Bim1 complex promotes Bik1 binding, which facilitates Dam1c ring formation independently of microtubules. Furthermore, they suggest that the Bim1-Duo1 interaction is suppressed by the Dam1 C-terminus but is promoted by Mps1 kinase. This is a comprehensive study revealing the Dam1c interaction with +TIP proteins, which gives important insights into how the Dam1c locates preferentially at the microtubule plus end to support the kinetochore interaction. Most experiments were carried out in high standard and the majority of results reasonably support their conclusions.

Major points:

Whereas the majority of results reasonably support their conclusions, the evidence for Mps1-dependent regulation is relatively weak in the following two points:

First, although robust data have been presented with Mps1 overexpression (Fig 4F, 4G, 4H, 4I and S4E), such gain-of-function data may not necessarily reflect physiological regulation - for example, potential phosphorylation of non-physiological substrates may be involved. I suggest strengthening loss-of-function data, which are not robust at present. For example, although Fig 4E uses an mps1 temperature-sensitive mutant, I wonder if pull-down of Duo1 is really reduced with this mutant at 37 degree, as the amount of GST-Bim1 also seems reduced in that condition. More quantitative data should be provided and reproducibility of the result should be confirmed. In addition, more data should be obtained with the mps1 temperature-sensitive, e.g. a change in Dad1-GFP localization.

Second, although the authors conclude that Duo1 N-terminus is an important substrate of Mps1 in regulation of Dam1c-Bim1 interaction, this conclusion relies only on a small change in Dad1-GFP signals with duo1-8A. More evidence should be provided to strengthen this conclusion - e.g. a change in the amount of Dam1c pulled down by GST-Bim1 when Dam1c contains Duo1-8A.

If it is technically difficult to strengthen their conclusion in these points, their conclusion about Mps1-dependent regulation should be toned down and can be removed from the title - I think the Dam1c regulation by Bim1 and Bik1 is a novel finding and the title and abstract can focus on it.

Optional point:

All electron microscopy results of Dam1c oligomerization and ring formation were obtained in the absence of microtubules in this study. However, Dam1c oligomerization and ring formation can take place on microtubules, which is functionally more important. I therefore suggest studying how Bim1 and Bik1 promote Dam1c oligomerization and ring formation on microtubules, using electron microscopy. However, this could be time-consuming and I understand that if this cannot be done as a part of revision of this manuscript.

Other specific points:

Suppl Fig 1A: Because Stu2 and Dam1c elute in the same fractions when they are analyzed separately, it is hard to tell whether Stu2-Dam1c interaction occurs when Stu2 and Dam1c are mixed. The authors should improve the resolution of chromatography or use another method to study Stu2-Dam1c interaction. However, if this is difficult, they can withdraw the data and conclusion about Stu2-Dam1c interaction, which is not a part of their main conclusion in this study.

Line 229-231: With the duo1 mutant lacking SxIP, it is concluded that 'a high number of large budded cells with abnormally large bud size and short inter-kinetochore distance was observed', based on Fig 3C. This conclusion should be substantiated by more quantitative analyses, rather than relying only on representative cell images. I also wonder if they meant to say 'large inter-kinetochore distance' rather than 'short inter-kinetochore distance'.

Fig 3E: The differences in both Pds1 levels and FACS DNA contents are relatively small between the wild-type control and the duo1 mutant lacking SxIP. I wonder if the experiment was repeated to confirm reproducibility of the result.

While the duo1 mutant lacking SxIP only marginally reduces cell viability (Fig 3A), this mutant plus Ask-2A reduces cell viability more clearly (Fig 7B). It will be interesting and important to evaluate the changes in Dad1-GFP localization and/or in metaphase length etc when the two mutants are combined.

Fig 7C: Both in vitro and in vivo data in literature indicate that the Dam1c ring is assembled on a microtubule independently of kinetochores. In addition, in the current study, Bim1-Dam1c binding is seen without Mps1 phosphorylation. Fig 7C (top) is not in line with these results, although the Dam1c ring formation may indeed be enhanced further by Mps1 kinase as shown in Fig 7C (bottom).

Minor points:

Line 27-28: Both in vitro and in vivo data in literature indicate that the Dam1c ring is assembled on a microtubule independently of kinetochores. Therefore, the sentence 'How ring assembly is specifically initiated at kinetochores....' is misleading.

Line 155-155: 'In the presence of Bim1, Dam1c was mainly found as a dimer': What fraction of Dam1c was found as a dimer?

Line 163: '...identified crosslinks between Bim1 and Spc19/Spc34 and Duo1 and Spc19/Spc34': This phrase is confusing. Perhaps 'between' can be inserted before 'Duo1'.

Line 277-278: It says '300 mM NaCl' in main text. However, it says '400 mM NaCl' in the Fig 4A legend. Which is correct?

Line 348-349: In the given PDF file, the 'elongated structures' are not visible in Fig 4G, bottom panel, in contract to the description in main text.

Line 683: 'und' should be 'and'

Line 744: 'Lysates were by centrifugation and incubated...'. I think a word is missing between 'were' and 'by'.

Line 1217-1218: 'SDS-PAGE analysis of Dam1c in the absence of Bim1-Bik1 are shown in Supplementary Figure 7B.' In this sentence, 'Supplementary Figure 7B' should be 'Supplementary Figure 6B'.

Line 1331-1332: 'Exemplary images of the respective categories are depicted in A'. In this sentence, 'depicted in A' should be 'depicted in B'.

Fig 1B: To distinguish blue and green lines, blue lines can be labelled as e.g. Intermolecular crosslinks 'between Dam1c components'.

Fig 3B: Please clarify whether the strains for this FACS analysis had Dad1-GFP or not.

Re: EMBOJ-2021-108004

Mps1 controlled +TIP interactions trigger Dam1c ring assembly at the outer kinetochore

Point-by-point response

Please find below our point-by-point answer to the reviewer's comments, with our answers in red lettering

Referee #1:

Dudziak et al use a combination of biochemistry, cross-linking mass spectrometry, single particle electron microscopy (EM), cell imaging, and genetic techniques to examine the interaction between the budding yeast +TIP, Bim1, and the Dam1c kinetochore complex. Dam1c is well-known as a component of the yeast outer kinetochore crucial for stabilizing attachments between chromosomes and microtubules in the mitotic spindle. In vitro, Dam1c oligomerizes into striking microtubule-encircling rings that enable kinetochores to make strong yet dynamic attachments to growing and shrinking microtubule ends. Bim1, like its homologue EB1, famously binds and tracks growing microtubule plus ends. A prior two-hybrid screen suggested that Bim1 and Dam1c might bind to one another but the relevance of their interaction was unexplored. This new study presents the first analysis of the subdomains involved in the Bim1-Dam1c interaction and provides an initial view of the structural arrangement. The data are high quality, novel, and provide support for an interesting potential role for the Bim1-Dam1c interaction as a means to orient the Dam1c for proper loading onto its kinetochore receptor, Ndc80c.

We thank the reviewer for this positive evaluation of our manuscript.

Unfortunately, however, the implications are overstated in at least two ways, listed below. In my opinion the authors need to deal with these overinterpretations before the manuscript is ready for publication.

It is asserted at least three times in the manuscript that the binding of Bim1 or Ndc80c to Dam1 is mutually exclusive. (E.g., in the section heading on line 422. Also on lines 440-441. Also on lines 656-657.) No data presented here can support this claim. A mere 30% reduction in the amount of Dam1c pulled down by immobilized Ndc80-FLAG was observed after addition of 16 μ M Bim1 (Figure 6B and C). This observation indicates that Bim1 weakly inhibits the Dam1c-Ndc80c interaction. A bulk pull-down experiment like this is insufficient to test for mutual exclusivity.

*We agree with the reviewer that our data regarding this point do not allow an unambiguous interpretation and we have rephrased the corresponding statements at the indicated positions. In the revised manuscript we have added additional experiments analyzing Dam1c binding to either Bim1-Bik1, or to Ndc80c in solution by SEC (**new Expanded View Figure 5 B and C**). These experiments indicate that similar to Bim1-Bik1, Ndc80c binds Dam1c in solution and both complexes are present in very early eluting fractions indicating the formation of high-molecular weight complexes under these conditions. When all three components were combined (Dam1c+Bim1-Bik1+Ndc80c), the amount of Ndc80 eluting early was not changed substantially relative to the sample Dam1c+Ndc80c. Therefore, we continue to favor the idea that Dam1c-Bim1-Bik1 and Dam1c-Ndc80c represent distinct complexes. We agree however, that answering this question will require more sophisticated*

methods - ideally structural characterization of the respective assemblies - in future studies. We have added a statement to this effect to the discussion.

It is also asserted that Mps1-dependent phosphorylation of Dam1c "increases the affinity of Dam1c for Bim1" (lines 519-520; a similar assertion is also made on lines 355-356). This conclusion is only indirectly supported, by decreased/increased pull-down of Dam1c from clarified lysates from cells with mutant/over-expressed Mps1, using an immobilized fragment of Bim1 as bait. Since Mps1 phosphorylates many targets in cells, these differences are not necessarily due to changes in the biochemical affinity of Dam1c for Bim1.

This is a valid point, also brought up by reviewer 3. We have tried to further address this issue in the revised manuscript in two ways:

1). Additional loss-of-function alleles (mps1-as1), confirm the findings with the temperature-sensitive allele in the pull-down assay (see answer to Reviewer 3).

*2) We have used direct in vitro binding assays with Mps1-phosphorylated components (**new Expanded View Figure 3 B-D**). We find two distinct effects of Mps1 phosphorylation in this reconstituted system: 1. Phosphorylation of Dam1c by Mps1 leads to a slightly later elution of Dam1c from the SEC, suggesting a shift from oligomers to monomers upon phosphorylation. 2. When tested for Bim1 binding under high salt concentrations (400 mM NaCl, similar to the Dam1-19 analysis), Mps1 phosphorylation led to a more complete co-elution of phosphorylated Dam1c with Bim1, similar to the effect of the Dam1-19 mutation (Figure 4). Note the shifted elution profile of Dam1c-Bim1 phosphorylated by Mps1, compared to the corresponding sample lacking Bim1. We consider this result as an indication that Mps1 phosphorylation can indeed directly promote binding of Bim1 to Dam1c in vitro and mimics the effect of the dam1-19 mutation. We concede that by comparison, the issue of Mps1 regulation is less well supported than other findings. We have therefore de-emphasized the Mps1 regulation aspect in title, abstract and text, in line with the suggestion made by reviewer 3.*

Referee #2:

The MS of Dudziak et al explores how plus-end associated proteins regulate the assembly of Dam1c as rings at attached kinetochores. A combination of biochemistry, genetics, and structural analysis are nicely combined to tease out the interactions between Bim1 (EB1 ortholog), Bik1, Dam1c, Ndc80c, and Mps1. It shows that Bim1 alone promotes oligomerization of Dam1c as partial rings and that Bim1 plus Bik1 promotes Dam1c's oligomerization into complete rings. This interaction is enhanced by Mps1, which, due to its localization at unattached kinetochores, suggests a model in which Mps1 facilitates Dam1 ring formation at the kinetochore microtubule plus end that has just attached. Through the identification of a possible Bim1:Dam1c interface, the MS also proposes that Bim1, which is localized to microtubule plus ends, can select for Dam1c that is oriented with its protrusion pointed toward the microtubule plus end. This model also explains how Dam1c, which by itself does not appear to sense microtubule polarity, is able to assemble with the kinetochore in the correct orientation. I think this MS contributes important advances to the kinetochore field and suggests exciting follow-up studies, particularly a high-resolution characterization of the newly reconstituted Bim1-Dam1c and Bim1-Bik1-Dam1c complexes.

We thank the referee for the positive review of our manuscript.

I only have minor concerns, which if addressed should make the MS clearer to a broader readership.

Line 43 "Faithful and error-free chromosome segregation" - The wording seems redundant. Isn't error-free segregation already faithful?

Agreed, we have changed this accordingly.

Line 126 "This result shows that Bim1 is a specific binding partner of Dam1c at microtubule plus ends" - While the experiments in this paragraph do show that Bim1 and Dam1c bind specifically, they don't show that they bind at the MT plus ends. To support such a claim, one would have to show that the complex forms on MT plus ends, either in vitro in situ. If this statement is based on light microscopy co-localization data, then the relevant papers should be cited and the strong term "shows" replaced with the more tenuous term "suggests".

We have removed "...at microtubule plus ends" from this sentence.

Line 157 "Bim1 was visible as an additional mass of approximately 150 kDa" - How was the mass of the putative Bim1 density estimated from the class average? What is the basis for the conclusion that the extra density at the protrusion domain is a Bim1 homodimer? There seems to be a circular argument going on here. Is there additional evidence from the data, which may not be obvious to non-structural-biology readers?

We have rephrased our wording in this section, as the resolution of the class averages does indeed not allow definitive statements regarding the number of Bim1 molecules and the stoichiometry. The section now reads: "Bim1 was visible as an additional mass crowning the protrusions domains of adjacent heterodecamers. The size of the extra mass is estimated to accommodate at least one homodimeric Bim1 molecule (76,6 kDa)".

Lines 170-174 "In contrast, the Dam1c-Bim1 complex showed larger oligomers than Dam1c alone, with a curvature that is consistent with early steps of ring formation." -This statement implies that the curvature of Dam1c multimers during early ring formation is known. Please clarify.

We agree that this sentence is not precisely worded and corrected it accordingly. By this statement we want to point to the fact that the curvature of the partial rings fits to the curvature of fully assembled rings. Thus, we conclude that the partial rings resemble early steps of ring formation.

Lines 145-174 - Please comment on the confusing relationship between the apparent masses of the various complexes, seen by gel filtration versus by EM. Gel filtration shows that Dam1c alone elutes as a mono-modal species larger than 670 kDa whereas the EM suggests a mixture of monomers and dimers. When Bim1 is added, Gel filtration still shows a monomodal species whereas the EM shows a shift to the Dam1c dimer. The Dam1c monomer is ~ 200 kDa and a Bim1 monomer is ~ 75 kDa, so a 2:2 complex of Dam1c:Bim1 would be 550 kDa. Should we conclude from the chromatogram in Fig. 1A that Dam1c alone in solution is actually a multimer and that when subjected to EM sample preparation, dissociates into monomers unless stabilized by Bim1?

Thank you for addressing this very interesting point. The oligomeric status of Dam1c, is influenced by multiple factors including protein concentration, ionic strength of the buffer, presence of binding partners, etc. Due to the extended conformation of even an individual T-

shaped Dam1c heterodecamer, the elution position in SEC is not suitable to determine molecular weight and oligomeric state. For the future, analytical ultracentrifugation, determining the sedimentation constant as a function of concentration etc., would be desirable. A comparison between SEC and EM is further complicated by the protein dilution necessary for sample preparation and the question if the sample is fixed or not prior to dilution. To evaluate the effect on the oligomeric state of Dam1c it is therefore important to only change one parameter. For example, the effect of Bim1 on the oligomeric state of Dam1c can best be appreciated in the comparison between Fig. 1E and 1F, as the only variable here is the inclusion of Bim1. We have tried to make these points more clear in the revised manuscript.

Lines 234 "Strikingly, the kinetochore clusters tumbled in the nucleus" - The nucleus is not labelled in this experiment, so one cannot exclude the possibility that it's the nucleus that's tumbling around inside the cell. A tumbling nucleus would hint at changes in the cytoplasmic portion of the cytoskeleton, in addition to the spindle.

Agreed, we corrected this statement accordingly.

Line 250: "substantial Pds1 were still detectable". Perhaps quantifying the western blots at Fig. 3E would help show the "substantial" difference in Pds1 levels for WT versus Duo1[delta]SxIP strain?

*Thank you for this helpful comment. We quantified the Pds1 signals and normalized it to the Pdk1 signal. The corresponding graph is shown in the **new Figure 3G**.*

Lines 347-349 and Fig. 4G (see below) I don't see the "elongated structures" indicated by the white arrows.

We apologize for this mistake. Unfortunately, images appear darker after conversion from the Adobe Illustrator file to PDF. We have corrected the brightness of this image in Figure 4G to make the elongated structure visible again.

Lines 406-407 Replace "degradation" with "dissociation".

We have changed the wording accordingly.

Lines 408-409 Replace "and have a relatively thick circumference" with "and are thicker than rings assembled in the presence of Bim1 alone"

We have rephrased the sentence accordingly.

Line 541 The notion of a metaphase-specific configuration of Dam1c is an interesting one and should be briefly discussed relative to an earlier study (Dhatchinamoorthy et al "Structural plasticity of the living kinetochore."), which showed evidence that Ndc80c has a anaphase configuration while Dam1c did not, as assessed by copy numbers in live-cell fluorescence imaging.

Thank you for this valuable comment. In our revised manuscript, we briefly discuss our results in context of the above-mentioned study.

Lines 635-636 "Thus, ring formation is prevented at any other site such as already attached kinetochores or microtubules distal from unattached kinetochores." - This sentence is poorly worded. It makes it sound like Dam1c rings are prevented from forming at the already attached kinetochores. It's probably safer to say that "ring-formation is not enhanced at any other site".

We revised the sentence and corrected it to "By this mechanism, Dam1c ring formation is exclusively restricted to unattached kinetochores where it is required for formation of stable

kinetochore-microtubule attachments, while it is not enhanced at any other site such as microtubules distal from unattached kinetochores."

Lines 646-656 "Since Bim1 preferentially binds the microtubule plus end, Dam1c would be positioned in such a manner that the protrusion domain points towards the kinetochore" - This statement makes assumptions about the orientation Bim1's Dam1c-binding domain when it's on the microtubule. It assumes that Bim1's Dam1c binding domain is oriented parallel to the MT surface, pointing toward the plus end. If, instead, Bim1's Dam1c domain is oriented perpendicular to the MT's surface, then the proposed model wouldn't make sense, and would require additional modifications such as a conformational change upon Dam1c binding. Is there any evidence that this interface is indeed oriented toward the minus end?

That's an interesting point. The conformation of a full length EB protein at a plus end is indeed not known, it would also be expected that there is some flexibility between CH domains and cargo domain, allowing different conformations. Our idea here is, simply by being the most plus-end proximal factor, Bim1 may aid not only in the local enrichment and oligomerization of Dam1c, but also could contribute to proper orientation. We have made clear in the discussion that this point is indeed speculative.

OTHER POINTS

Please clarify what "+TIP" acronym means, for non-expert readers.

We added a brief explanation of the term in the introductory part.

The term "Negative stain EM images of X" is inaccurate. It should be "EM images of negatively stained X".

We have corrected this at the appropriate positions in the legends for figures 1, 5 and EV1.

The phrase "Interestingly/strikingly ..." is used too often. It implies surprise at a newly presented result when in most cases, the result was fully consistent with those that came before the term interestingly/strikingly. If "Consistent / In agreement with the previous result ..." is substituted for "strikingly", I believe the MS will make more sense. If the results are indeed striking/interesting, please explain why.

Thank you for this recommendation. We changed the corresponding passages whenever appropriate.

The MS reports that overexpression of Mps1 affects the localization of Dad1-GFP in metaphase. This is characterised by the increase in percentage of cells showing bar-shaped Dad1-GFP localisation (Fig. 4G, H, I). Could the formation of this bar-shaped morphology simply be the result of excess accumulation Dad1-GFP / Dam1c? Fig. 4H seems to indicate that the overall Dad1-GFP intensity is higher when Mps1 was overexpressed.

We agree that the bar-shaped signal might be a result of excess accumulation of Dad1-GFP, which may not only reflect kinetochores, but also other microtubule plus-ends in the spindle. The small size of the yeast metaphase spindle does not allow to distinguish individual plus-ends or different types of microtubules by light microscopy. Overall, this observation supports the notion that increased Mps1 signaling enhances Dam1c association to kinetochores/spindles.

Lines 638-85 + Fig. 5D,E + Fig. 7C: The in vitro experiments show that it is possible to assemble complete rings without microtubules. Please speculate on what this finding means in vivo: Is it possible that inside cells, complete rings form first and then thread on the

kinetochore microtubules? Or are their conditions that ensure ring formation only happens in the context of microtubules? I believe this MS provides enough evidence against the former possibility, due to the localization of Bim1 on microtubules in vivo. Though one cannot rule that there are sufficient copies of soluble Bim1, Bik1, and Dam1c to form rings in nucleoplasm, independently of the spindle.

Thank you for raising this interesting point. We think the ability to form full rings without microtubules in the presence of Bim1-Bik1 highlights the activity that these proteins have regarding Dam1c oligomerization. We think it's very likely that in vivo this process only occurs in the presence of microtubules. We added a brief discussion of this possibility in our revised manuscript.

FIGURES

In the chromatogram of Fig. S1A, the positions of the Dam1c (alone) and Stu2 (alone) peaks are awfully close. Is there a co-IP experiment to better support the conclusion that Stud2 and Dam1c don't interact in solution?

We thank the reviewer for this suggestion. To further validate our finding, we performed a pull down assay with Stu2-Flag immobilized on beads. We added recombinant Dam1c and Bik1 as positive control. The result supports our previous conclusion that Stu2 does not bind to Dam1c (new Expanded View Figure 1B).

The legends for Fig. 1A and S1A don't say what the gels correspond to. Based on the color scheme, they are presumably fractions collected from the chromatography runs. It would be good to say so in the legend.

We re-checked this: the corresponding figure legends already contain the relevant information.

Fig. 1D: I suggest replacing the blue and yellow ovals with dashed oval lines. The current rendering makes it look like the relevant features are behind the density map.

We have done this in the revised manuscript.

Fig. 3E: Why was the Dad1-GFP background strain used for the cell cycle progression experiments? In line 221, it's stated that the Dad1-GFP + Duo1[delta]SxlP suffer severe growth defects at 37C, compared to strain that expresses untagged Dad1n. I think doing these experiments (Pds1 western blots and FACS) on an untagged Dad1 background strain would better support the conclusion at line 525 that Duo1[delta]SxlP allele causes delayed cell cycle progression.

In Fig 3E we decided to use the Dad1-GFP strain background since this additionally pronounces the growth defect caused by the Duo1^{ΔSxlP} allele. Since both Duo1^{WT} and Duo1^{ΔSxlP} only differ in the Duo1 allele, one can clearly attribute the effect seen in our experiment to the Duo1^{ΔSxlP} allele which is deficient in Bim1 binding. Our serial dilution assay demonstrates that growth is only weakly compromised in the Duo1^{ΔSxlP} strain with untagged Dad1. Thus, we did not expect to see a significant difference when analyzing progression through a single cell cycle. Note, however, that Fig. 3B uses a wild-type background to report a Duo1deltaSxlP effect on the distribution of 2C DNA cells in unsynchronized cells.

Fig. 4G, lower panels: the arrowheads don't seem to be pointing to anything.

We apologize for this mistake. Unfortunately, images appear darker after conversion from the Adobe Illustrator file to PDF. We corrected the brightness of this image to make the elongated structure visible.

Fig. S4E - In the pGal-Mps1 cell, the nucleus appears bigger and the Nup signals appear more dispersed. Could this difference be a side-effect of Mps1 overexpression?

We think that the change in the Nup60 appearance is indeed a side effect of the metaphase arrest caused by Mps1 overexpression. Presumably, the nucleus increases in size while the cell is arrested in metaphase leading to a more dispersed distribution of Nup60.

Lu Gan

Referee #3:

In this study, the authors found that yeast +TIP protein Bim1 directly binds the Dam1 complex (Dam1c). This binding is mediated by the SxIP motif at the Duo1 C-terminus. With the duo1 mutant lacking SxIP, the amount of the Dam1c is reduced at kinetochores in yeast cells and metaphase is prolonged. Moreover, the Dam1c-Bim1 complex promotes Bik1 binding, which facilitates Dam1c ring formation independently of microtubules. Furthermore, they suggest that the Bim1-Duo1 interaction is suppressed by the Dam1 C-terminus but is promoted by Mps1 kinase. This is a comprehensive study revealing the Dam1c interaction with +TIP proteins, which gives important insights into how the Dam1c locates preferentially at the microtubule plus end to support the kinetochore interaction. Most experiments were carried out in high standard and the majority of results reasonably support their conclusions.

We thank the referee for the positive evaluation of our study.

Major points:

Whereas the majority of results reasonably support their conclusions, the evidence for Mps1-dependent regulation is relatively weak in the following two points:

First, although robust data have been presented with Mps1 overexpression (Fig 4F, 4G, 4H, 4I and S4E), such gain-of-function data may not necessarily reflect physiological regulation - for example, potential phosphorylation of non-physiological substrates may be involved. I suggest strengthening loss-of-function data, which are not robust at present. For example, although Fig 4E uses an mps1 temperature-sensitive mutant, I wonder if pull-down of Duo1 is really reduced with this mutant at 37 degree, as the amount of GST-Bim1 also seems reduced in that condition. More quantitative data should be provided and reproducibility of the result should be confirmed. In addition, more data should be obtained with the mps1 temperature-sensitive, e.g. a change in Dad1-GFP localization.

Thank you for addressing these aspects to improve our manuscript. We agree that the effect of Mps1 overexpression is difficult to interpret since phosphorylation of other substrates than Dam1c might contribute to the observed phenotype. However, we have similar concerns regarding experiments involving Mps1 inhibition since Mps1 activity or inhibition is also closely linked to Ipl1/Aurora B and phosphatase activity at the kinetochore (reviewed in Saurin, 2018).

Our pull down data is reproducible (see new Appendix Figure S3), also with an analog-sensitive Mps1 allele (mps1-as1).

We analyzed Dad1-GFP localization under Mps1 inhibition. Since Mps1 is essential for

spindle pole body duplication and its inhibition results in monopolar spindles, we decided to acutely inhibit Mps1 by the small molecule cincreasin (Dorer et al., 2005) which inhibits Mps1's function at the kinetochore but still allows spindle pole body duplication. Our live cell microscopy data revealed a significant reduction in Dad1-GFP signal intensity at metaphase kinetochore clusters after Mps1 inhibition compared to control treated cells. Furthermore, we observed a similar effect after acute inhibition of analog-sensitive Mps1 with 1NM-PP1 (see figure below, presented as new Appendix Figure S3). We consider this data as an additional indication that Mps1 kinase activity promotes localization of Dam1c to kinetochores.

Effect of Mps1 inhibition on Dam1c-Bim1 interaction and Dam1c kinetochore localization

A: Pull down assay with soluble cell lysates to analyze Dam1c binding to Bim1 after selective inhibition of either Mps1 (*mps1-as1*) or Ipl1 (*ipl1-2*) or both. Mps1 was inhibited by addition of the ATP analog 1NM-PP1, Ipl1 by growing cells at the restrictive temperature of 37 °C. A strain with wild type Mps1 and Ipl1 was used as control.

B: Quantification of Dad1-GFP signal intensities at metaphase kinetochore clusters after treatment with 0.5 mM or 1 mM cincreasin for 2.5 hours. A control sample was treated with 0.1 % (v/v) DMSO.

C: Quantification of Dad1-GFP signal intensities at metaphase kinetochore clusters. Cells carrying the *mps1-as1* allele were either treated with 1 % DMSO or 10 μM 1NM-PP1 for 10 or 30 minutes.

B and C: n ≥ 98 clusters per condition, p-values were calculated by a one-way ANOVA test with Dunnett's test for multiple comparisons

Second, although the authors conclude that Duo1 N-terminus is an important substrate of Mps1 in regulation of Dam1c-Bim1 interaction, this conclusion relies only on a small change in Dad1-GFP signals with duo1-8A. More evidence should be provided to strengthen this conclusion - e.g. a change in the amount of Dam1c pulled down by GST-Bim1 when Dam1c contains Duo1-8A.

We agree that the effect of the Duo1^{8A} allele observed in the serial dilution assay and live cell microscopy is relatively weak. Our mass spec analysis of phosphorylation sites probably missed physiologically relevant phospho-sites. Thus, we decided to remove these data from our manuscript and will further investigate the relevant phosphorylation sites within the Dam1 complex in future studies.

If it is technically difficult to strengthen their conclusion in these points, their conclusion about Mps1-dependent regulation should be toned down and can be removed from the title - I think the Dam1c regulation by Bim1 and Bik1 is a novel finding and the title and abstract can focus on it.

We have de-emphasized our conclusions regarding the Mps1-dependent regulation in title, abstract and in the text.

Optional point:

All electron microscopy results of Dam1c oligomerization and ring formation were obtained in the absence of microtubules in this study. However, Dam1c oligomerization and ring formation can take place on microtubules, which is functionally more important. I therefore suggest studying how Bim1 and Bik1 promote Dam1c oligomerization and ring formation on microtubules, using electron microscopy. However, this could be time-consuming and I understand that if this cannot be done as a part of revision of this manuscript.

This is definitely an interesting point, and we plan to analyze the complex assembled on microtubules in future studies.

Other specific points:

Suppl Fig 1A: Because Stu2 and Dam1c elute in the same fractions when they are analyzed separately, it is hard to tell whether Stu2-Dam1c interaction occurs when Stu2 and Dam1c are mixed. The authors should improve the resolution of chromatography or use another method to study Stu2-Dam1c interaction. However, if this is difficult, they can withdraw the data and conclusion about Stu2-Dam1c interaction, which is not a part of their main conclusion in this study.

*Thank you for mentioning this issue (see also Reviewer 2). We analyzed binding of Dam1c to immobilized Stu2 on beads and found that Dam1c does not interact with Stu2 (**new Figure EV1B**). We included Bik1 as a positive control under the same conditions, showing that immobilized Stu2 is capable of interacting with genuine binding partners.*

Line 229-231: With the duo1 mutant lacking SxIP, it is concluded that 'a high number of large budded cells with abnormally large bud size and short inter-kinetochore distance was observed', based on Fig 3C. This conclusion should be substantiated by more quantitative analyses, rather than relying only on representative cell images. I also wonder if they meant to say 'large inter-kinetochore distance' rather than 'short inter-kinetochore distance'.

In the revised manuscript, we quantified the proportion of cells with almost equally sized bud and mother cell (“very-large” or “XL-budded”) having a short inter-kinetochore distance and added the corresponding graph to our revised manuscript (new Figure 3E). At this point we intended to say “short inter-kinetochore distance” since a short spindle is characteristic of metaphase cells.

Fig 3E: The differences in both Pds1 levels and FACS DNA contents are relatively small between the wild-type control and the duo1 mutant lacking SxIP. I wonder if the experiment was repeated to confirm reproducibility of the result.

These experiments were additionally performed at 30°C with a similar result. To substantiate our findings, we have added a quantification of the Pds1 level (new Figure 3G, see answer to Reviewer 2).

While the duo1 mutant lacking SxIP only marginally reduces cell viability (Fig 3A), this mutant plus Ask-2A reduces cell viability more clearly (Fig 7B). It will be interesting and important to evaluate the changes in Dad1-GFP localization and/or in metaphase length etc when the two mutants are combined.

The genetic interaction between Duo1^{ΔSxIP} and Ask1^{2A} is an interesting and important aspect that we would like to address in the future. So far, we can report that combining both Duo1^{ΔSxIP} and Ask1^{2A} alleles does not further reduce Dam1c levels at metaphase kinetochores compared to the Duo1^{ΔSxIP} allele alone, as judged by Dad1-GFP fluorescence intensity. Possibly, the synthetic effect is caused by a more complicated mechanism and requires more detailed analysis in future.

Fig 7C: Both in vitro and in vivo data in literature indicate that the Dam1c ring is assembled on a microtubule independently of kinetochores. In addition, in the current study, Bim1-Dam1c binding is seen without Mps1 phosphorylation. Fig 7C (top) is not in line with these results, although the Dam1c ring formation may indeed be enhanced further by Mps1 kinase as shown in Fig 7C (bottom).

Regarding Dam1c ring assembly in vivo and in vitro we refer to our response to the next point.

It is true that we report Bim1 binding to Dam1c even without Mps1 phosphorylation. However, we think that Mps1-dependent phosphorylation does not function as an on-and-off switch that either allows or prevents binding, but rather gradually affects Dam1c's affinity towards Bim1. As demonstrated by us and others, oligomerization of Dam1c is most likely regulated by multiple factors and phosphorylation of Dam1c by Mps1 and subsequent binding of Bim1 is only one of several triggers of oligomerization.

Minor points:

Line 27-28: Both in vitro and in vivo data in literature indicate that the Dam1c ring is assembled on a microtubule independently of kinetochores. Therefore, the sentence 'How ring assembly is specifically initiated at kinetochores....' is misleading.

We have changed this in the abstract to “...how ring assembly is initiated in vivo...”

Line 155-155: 'In the presence of Bim1, Dam1c was mainly found as a dimer': What fraction of Dam1c was found as a dimer?

We have corrected this statement in the revised manuscript. Supplementary Figures 1D and 1E (now Figures EV1D and E) are not directly comparable regarding the oligomerization status of Dam1c. The Dam1c-Bim1 sample was crosslinked to stabilize the complex during

EM sample preparation. In contrast, the Dam1c sample without Bim1 was not crosslinked. However, the samples shown in Figure 1E and F were processed identically which allows to draw reliable conclusions about Bim1's effect on Dam1c oligomerization. The EM analysis of these samples shows that binding of Bim1 promotes oligomerization of Dam1c or stabilizes its oligomeric forms. A quantification of the oligomerization status (e.g., distribution between monomer, dimer, trimer...) is not possible with the present data set. We have corrected the corresponding sentences in our revised manuscript.

Line 163: '...identified crosslinks between Bim1 and Spc19/Spc34 and Duo1 and Spc19/Spc34': This phrase is confusing. Perhaps 'between' can be inserted before 'Duo1'.

We changed the sentence to “crosslinks between Bim1 and Spc19/Spc34 and between Duo1 and Spc19/Spc34”.

Line 277-278: It says '300 mM NaCl' in main text. However, it says '400 mM NaCl' in the Fig 4A legend. Which is correct?

We apologize for this mistake. 400 mM NaCl, as stated in the figure legend is correct. We changed the corresponding passage in our manuscript.

Line 348-349: In the given PDF file, the 'elongated structures' are not visible in Fig 4G, bottom panel, in contract to the description in main text.

We apologize for this mistake. The brightness of the images was changed during conversion of the Adobe Illustrator file to PDF. We adjusted the brightness of the image to make the elongated structure clearly visible.

Line 683: 'und' should be 'and'

We corrected this very German typo in our revised manuscript.

Line 744: 'Lysates were by centrifugation and incubated...'. I think a word is missing between 'were' and 'by'.

We corrected the sentence to “Lysates were cleared by centrifugation and incubated...”

Line 1217-1218: 'SDS-PAGE analysis of Dam1c in the absence of Bim1-Bik1 are shown in Supplementary Figure 7B.' In this sentence, 'Supplementary Figure 7B' should be 'Supplementary Figure 6B'.

Thank you. We corrected this mistake accordingly.

Line 1331-1332: 'Exemplary images of the respective categories are depicted in A'. In this sentence, 'depicted in A' should be 'depicted in B'.

Thank you for pointing out this mistake. We corrected the corresponding sentence.

Fig 1B: To distinguish blue and green lines, blue lines can be labelled as e.g. Intermolecular crosslinks 'between Dam1c components'.

We extended the figure labeling to “Intermolecular crosslinks between Dam1c subunits” and “Intermolecular crosslinks between Dam1c and Bim1”, respectively.

Fig 3B: Please clarify whether the strains for this FACS analysis had Dad1-GFP or not.

The strains for FACS analysis in Figure 3B contained untagged Dad1. We added a corresponding remark to the figure legend.

Thank you for submitting your revised manuscript to The EMBO Journal. Two of the original referees have now once more looked at it, and found the previously-raised points satisfactorily addressed. We shall therefore be happy to accept the study for publication in our journal, pending incorporation of the remaining minor referee comment (below) and the following editorial points:

Referee #2:

The authors have added many text clarifications, have done new experiments, and revised some of the figures. These changes have addressed my concerns. In my opinion, the MS is ready for publication. Congratulations!

Referee #3:

The authors addressed the majority of my points, including the major ones. One remaining issue is that, regarding the first minor point (Line 27-28: Both in vitro and in vivo data in literature...), it does not seem that the change mentioned in the authors' response is implemented in the abstract of the revised manuscript. Once this is implemented, I think the revised manuscript is ready for publication.

Referee #2:

The authors have added many text clarifications, have done new experiments, and revised some of the figures. These changes have addressed my concerns. In my opinion, the MS is ready for publication. Congratulations!

Thank you for the positive feedback on our manuscript.

Referee #3:

The authors addressed the majority of my points, including the major ones. One remaining issue is that, regarding the first minor point (Line 27-28: Both in vitro and in vivo data in literature....), it does not seem that the change mentioned in the authors' response is implemented in the abstract of the revised manuscript. Once this is implemented, I think the revised manuscript is ready for publication.

We apologize that we missed correcting this sentence accordingly. We now changed it to "...how ring formation is specifically initiated in vivo...".

At this point, we wish to thank all referees for their feedback and positive criticism that helped improve this study.

Thank you for submitting your final revised manuscript for our consideration. I am pleased to inform you that we have now accepted it for publication in The EMBO Journal.

Corresponding Author Name: Stefan Westermann

Manuscript Number: